# To View Transform or Not to View Transform: NeRF-based Pre-training Perspective

**Hyeonjun Jeong, Juyeb Shin, Dongsuk Kum**
KAIST, Daejeon, Korea
{hyeonjun.jeong, juyebshin, dskum}@kaist.ac.kr

## Abstract

Neural radiance fields (NeRFs) have emerged as a prominent pre-training paradigm for vision-centric autonomous driving, which enhances 3D geometry and appearance understanding in a fully self-supervised manner. To apply NeRF-based pre-training to 3D perception models, recent approaches have simply applied NeRFs to volumetric features obtained from view transformation. However, coupling NeRFs with view transformation inherits conflicting priors; view transformation imposes discrete and rigid representations, whereas radiance fields assume continuous and adaptive functions. When these opposing assumptions are forced into a single pipeline, the misalignment surfaces as blurry and ambiguous 3D representations that ultimately limit 3D scene understanding. Moreover, the NeRF network for pre-training is discarded during downstream tasks, resulting in inefficient utilization of enhanced 3D representations through NeRF. In this paper, we propose a novel NeRF-Resembled Point-based 3D detector that can learn continuous 3D representation and thus avoid the misaligned priors from view transformation. NeRP3D preserves the pre-trained NeRF network regardless of the tasks, inheriting the principle of continuous 3D representation learning and leading to greater potentials for both scene reconstruction and detection tasks. Experiments on nuScenes dataset demonstrate that our proposed approach significantly improves previous state-of-the-art methods, outperforming not only pretext scene reconstruction tasks but also downstream detection tasks.

## 1 Introduction

Accurate and fine-grained 3D scene understanding is essential for autonomous driving, supporting critical tasks such as 3D object detection (Reading et al., 2021; Li et al., 2023; 2024), high-definition (HD) map construction (Liao et al., 2023; Shin et al., 2025), and occupancy prediction (Tong et al., 2023; Tian et al., 2023). To facilitate these open-world perceptions, view transformation backbones (Li et al., 2023; 2024; 2022) have drawn great attention, which project multi-view 2D image features into a unified 3D representation on bird's-eye-view (BEV) or voxel space. A unified 3D representation, aligning various modalities (Liu et al., 2023b; Li et al., 2022; Yan et al., 2023; Kim et al., 2023) in a common metric frame, provides a single 3D canvas that can be leveraged across diverse downstream tasks (Hu et al., 2023; Jiang et al., 2023; Weng et al., 2024).

In parallel, neural fields, such as NeRFs (Mildenhall et al., 2021) and 3DGS (Kerbl et al., 2023), have emerged as a dominant paradigm for reconstructing 3D representation and synthesizing novel views by learning a continuous field of color and volume density in a self-supervised manner. Sharing the goal of understanding the 3D environment, recent studies (Yang et al., 2024; Huang et al., 2024; Xu et al., 2024) proposed combining NeRFs or 3DGS with view transformation, enabling self-supervised pre-training through photometric and depth reconstruction without the need for expensive manual annotations.

Although both view transformation and NeRFs ultimately aim to reconstruct a 3D representation of the world from 2D signals, they embody conflicting priors. Existing approaches (Yang et al., 2024; Huang et al., 2024) extract point features for radiance fields by interpolating discretized and fixed voxel features from a view transformation backbone, and then pre-train the backbone through photometric and depth errors rendered from those point features. However, this pipeline inevitably leads to NeRF inheriting the discrete and rigid priors of the view transformation, which conflicts

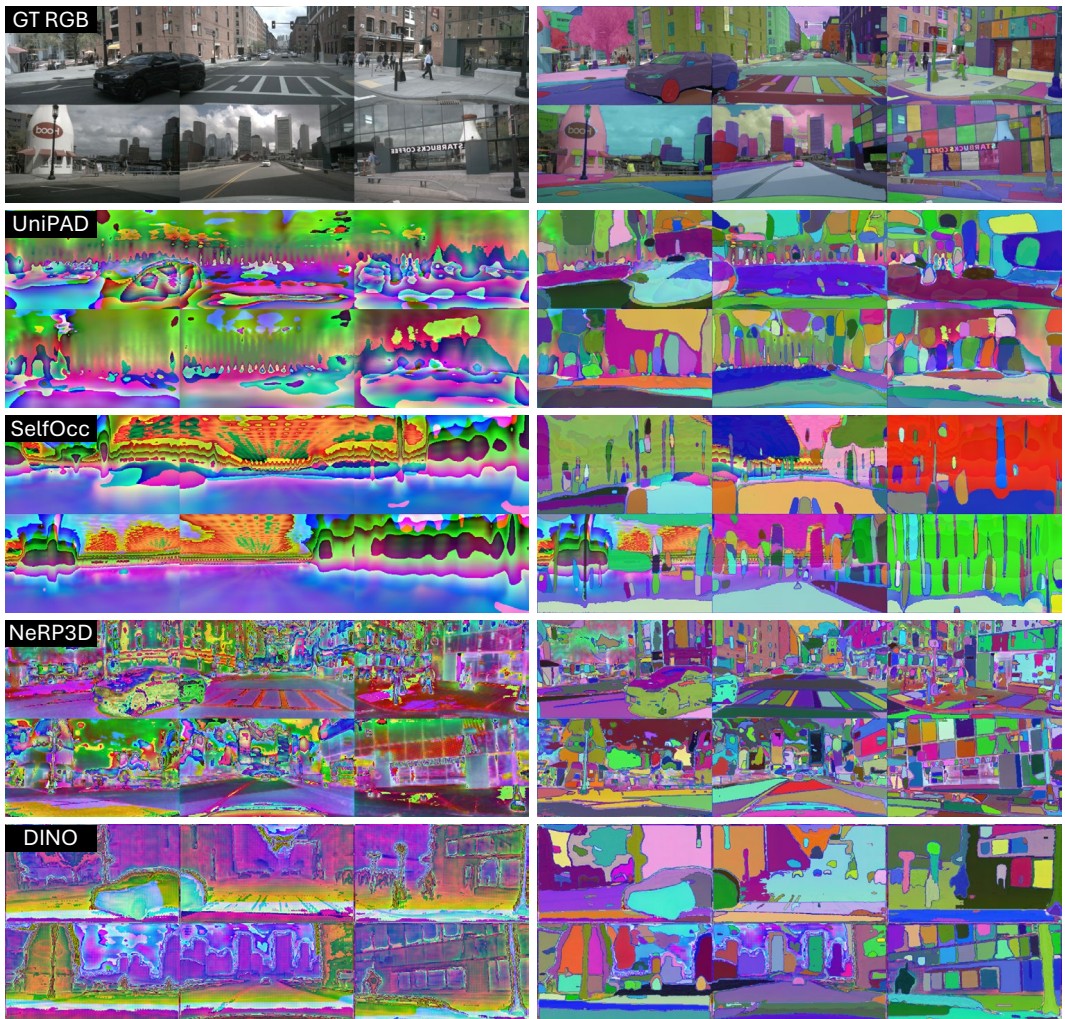

Figure 1: Comparison of 2D feature maps (left) and their instance segmentation (right) results using SAM (Kirillov et al., 2023; Ren et al., 2024; Ravi et al., 2024) across different methods. All 2D feature maps, except for ground truth RGB (row 1) and DINO (Caron et al., 2021; Oquab et al., 2023) feature (row 5), are obtained by accumulating 3D point-wise representations along each ray onto the image plane with predicted density. They are extracted directly after radiance field pre-training without any task-specific fine-tuning. UniPAD (Yang et al., 2024) (row 2) and SelfOcc (Huang et al., 2024) (row 3) produce blurry and inaccurate features that fail to separate nearby or crowded objects, resulting in under-segmented instances. In contrast, NeRP3D (row 4) produces precise and well-localized features with distinct object boundaries without any distillation or fine-tuning from 2D foundation models, comparable to those from DINO features. Consequently, we observe the potential for the enhancement of 3D representation to be reflected in the improved segmentation quality.

with the continuous radiance fields and restricts the fidelity of the reconstructed 3D representation. Moreover, the pre-trained NeRF is discarded during downstream tasks, preventing effective transfer of NeRF knowledge and limiting the exploitation of enhanced 3D representations from pre-training. As a result, distinct objects can be collapsed into a single blurry blob, as shown in Fig. 1.

In this paper, we introduce NeRP3D, a novel NeRF-Resembled Point-based 3D detector that fully inherits the continuous function of neural radiance fields (Mildenhall et al., 2021; Wang et al., 2021), effectively overcoming the inherent discrepancy with view transformation.Unlike methods relying on rigidly discretized voxel-based representations, NeRP3D directly models 3D scenes as continuous 3D features, geometry, and appearance from any continuous 3D location in a feedforward manner, as illustrated in Fig. 2. Experiments on the nuScenes (Caesar et al., 2020) benchmark demonstrate that

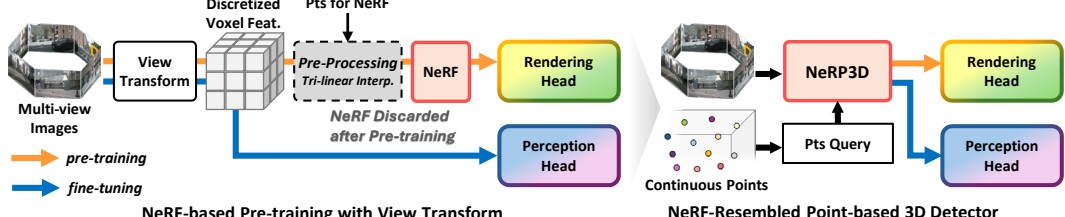

Figure 2: Comparison of the previous NeRF-based pre-training methods and our NeRP3D pipeline.

our approach significantly improves not only the rendering quality but also the downstream perception tasks for autonomous driving compared to previous approaches that simply incorporate NeRF-based pre-training into view transformation frameworks. These findings highlight the importance of aligning the 3D backbone with the pre-training model as well as continuous 3D representation learning in advancing NeRF-based pre-training for enhanced 3D scene understanding.

In summary, our contributions are:

- NeRP3D preserves the full knowledge from pre-training, since the NeRF-resembled design makes it effectively inherit and utilize continuous and fine-grained representations for both pretext and downstream tasks.

- Regardless of tasks, NeRP3D provides a unified framework allowing for consistent feature extraction with adaptive sampling, ray-wise and uniform spatial sampling, available through our proposed continuous function.

## 2 RELATED WORK

**Neural Radiance Fields** Neural radiance fields (NeRFs) (Mildenhall et al., 2021) and their variants (Wang et al., 2021; Fridovich-Keil et al., 2022; Müller et al., 2022; Barron et al., 2022; 2023) have established a powerful paradigm for 3D scene reconstruction by learning continuous volumetric functions from posed multi-view images. NeRFs are typically trained in a self-supervised manner, minimizing photometric reconstruction loss across multiple views. These prior works have demonstrated their ability to understand and enhance fine 3D geometry and appearance through high-fidelity novel view synthesis and 3D reconstruction. To move from dense toward sparse image sets, conditioning the radiance fields with image features (Yu et al., 2021; Chen et al., 2021; Liu et al., 2022b) shows reliable novel view synthesis results, demonstrating that generic 2D representations can guide NeRF training. Moreover, depth supervision (Roessle et al., 2022; Wei et al., 2023a; Deng et al., 2022; Wei et al., 2021) is incorporated to understand more accurate geometry. NeRF's enhanced 3D understanding is increasingly being extended to autonomous driving applications, and NeRP3D aims to fully leverage these capabilities.

**Neural Radiance Fields with Autonomous Driving** The inherent ability of NeRFs (Mildenhall et al., 2021; Wang et al., 2021; Barron et al., 2022; 2023) to capture 3D scene structure from multi-view 2D observations in a self-supervised manner has positioned them as a promising foundation for various autonomous driving applications. For sensor simulation in driving environments, offline scene reconstruction methods (Yang et al., 2023c; Tonderski et al., 2024; Yang et al., 2023b) have demonstrated NeRFs' capability to synthesize realistic camera images, generate scenarios through object manipulation, and decompose static-dynamic scenes. Moreover, DistillNeRF (Wang et al., 2024) builds upon EmerNeRF (Yang et al., 2023b) by extending it into a feed-forward model, while feature distillation from 2D foundation models (Radford et al., 2021; Oquab et al., 2023) further enhances 3D scene understanding.

The most relevant branch of this paper is the integration of NeRFs in pre-training to improve downstream perception tasks. UniPAD (Yang et al., 2024) introduces a universal NeRF-based pre-training framework to enhance the 3D object detection downstream task. Occupancy predictions (Huang et al., 2024; Zhang et al., 2023) are also integrated with NeRF, which is optimized through multi-view consistency (Godard et al., 2017; 2019; Zhou et al., 2017). GaussianPretrain (Xu et al., 2024) has demonstrated the feasibility of 3D Gaussian Splatting (Kerbl et al., 2023) for pre-training 3D scene

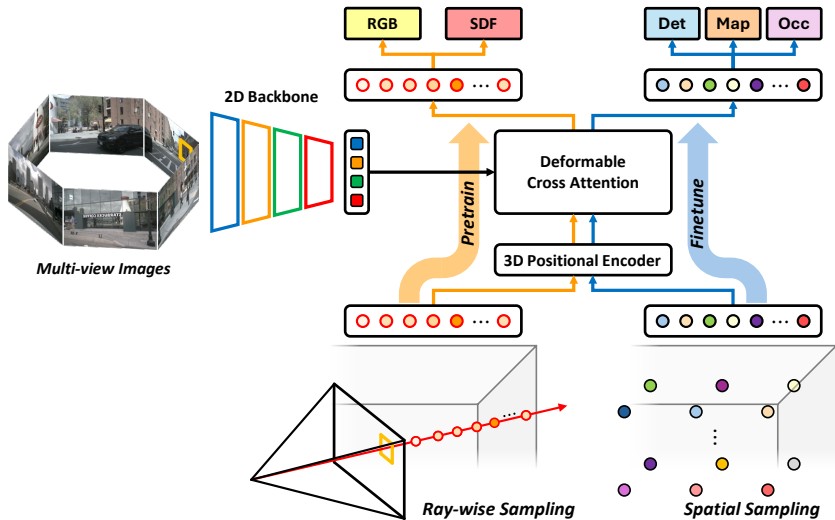

Figure 3: Overview of NeRP3D, illustrating both pre-training for rendering (orange) and fine-tuning for downstream (blue) pipelines. Through NeRF-resembled design, our method maintains a coherent 3D understanding from scattered points across diverse tasks while accommodating task-specific point sampling strategies, enabling the model to effectively leverage underlying geometric and appearance information while allowing for task-dependent feature specialization.

representations in driving environments. However, existing methods (Yang et al., 2024; Huang et al., 2024; Tian et al., 2023; Xu et al., 2024), which rely on view transformation, have inherent constraints that diminish NeRF's capacity for continuous and fine-grained 3D representation. Moreover, pre-trained NeRF is discarded during downstream tasks, resulting in suboptimal 3D representations enhancement from pre-training. In contrast, NeRP3D fully inherits pre-trained NeRF knowledge and utilizes continuous and fine-grained representations through its NeRF-resembled design.

## 3 METHOD

NeRP3D is a simple and effective NeRF-resembled architecture that unifies scene reconstruction and perception tasks from single-timestep multi-view images. As illustrated in Fig. 3, our framework operates in two distinct stages within a unified architecture, without discarding or adding modules depending on stage or task requirements. This unified architecture enables adaptive exploration of regions of interest tailored to specific processing efficiency, while maintaining a coherent 3D understanding across diverse tasks.

### 3.1 ADAPTIVE SAMPLING & REPRESENTATION OF POINT

To reconstruct accurate 3D representations from sparse and dynamic multi-view inputs, NeRP3D directly samples 3D points of interest at arbitrary spatial locations and predicts the representation of sampled points with 2D image features to cope with dynamic driving scenes, without processing voxelized feature grids or any interpolation from them.

NeRP3D first samples 3D points $\mathbf{x} \in \mathbb{R}^3$ using one of two distinct strategies tailored to different processing phases, view-dependent ray-wise sampling and uniform spatial sampling. For volumetric rendering, we follow the standard NeRF. Specifically, for each pixel in the multi-view images, we define a camera ray $\mathbf{r}_i$ based on its origin $\mathbf{o}_i$ and direction $\mathbf{d}_i$, which are derived from camera intrinsics and extrinsics. Along each ray, we sample a set of points $\{\mathbf{x}_{ij} = \mathbf{o}_i + t_j \mathbf{d}_i\}$ at regular or stratified distances within a defined range $\{t_j | j = 1, ..., D, t_j < t_{j+1}\}$. These sampled points are then integrated into rendered color and depth along the ray for differentiable volumetric rendering. In contrast, for downstream tasks, where the goal is to utilize the learned 3D representation for autonomous driving tasks such as 3D object detection or occupancy prediction, we sample points across the scene volume rather than following camera rays. We sample points $\mathbf{x}_{xyz}$ uniformly in 3D space around the vehicle, covering regions relevant to perception tasks.

Despite the difference in sampling methods, all 3D points, whether sampled along camera rays or spatially, are represented in the identical system, ensuring consistency across tasks and sharing a unified spatial understanding. In addition, we parameterize 3D coordinates to account for unbounded environments, inspired by Barron et al. (2022):

$$p(\mathbf{x}') = \begin{cases} \alpha \mathbf{x}' & |\mathbf{x}'| \leq 1 \\ \left(1 - \frac{(1-\alpha)}{|\mathbf{x}'|}\right) \frac{\mathbf{x}'}{|\mathbf{x}'|} & |\mathbf{x}'| > 1 \end{cases}, \tag{1}$$

where $p(\cdot)$ denotes a parameterized function that preserves real-scale coordinates for points within the inner range, while distributing distant points proportionally to disparity, including those at infinite distance. $\mathbf{x}'$ denotes normalized $\mathbf{x}$ to the range $[0, 1]$ and $\alpha \in [0, 1]$ denotes the contraction ratio.

After sampling 3D points, a set of 3D points $\{\mathbf{x}\}$ is conditioned with sparse 2D observations to represent 3D dynamic environments in a feed-forward manner. Given $N$ multi-view images $\{I_i\}_{i=1}^N$, we feed each image to the image backbone to obtain 2D image features $\mathbf{F} \in \mathbb{R}^{N \times H \times W \times C}$. Then, to enhance 3D point representations with image-aligned context, we adopt a deformable cross-attention (Zhu et al., 2020) with 2D image features $\mathbf{F}$. We first encode each 3D query point $\mathbf{x}$ by $\gamma(\cdot)$ and learn a set of $N_s$ sampling offsets $\{\Delta \pi_s \mid s = 1, ..., N_s\}$ relative to its projected 2D location $\pi(\mathbf{x})$, focusing interaction with relevant image regions. The final representation $\mathbf{z}$ of 3D point $\mathbf{x}$ is defined as:

$$\mathbf{z} = \sum_{h=1}^{N_h} \mathbf{W}_h \sum_{s=1}^{N_s} \mathbf{A}_{h,s} \mathbf{W}'_s \mathbf{F}(\pi(\mathbf{x}) + \Delta \pi_{h,s}(\gamma(p(\mathbf{x}')))), \tag{2}$$

where $\mathbf{N_h}$ denotes the number of heads for multi-head attention. $\mathbf{W_h} \in \mathbb{R}^{C \times (C/N_h)}$ and $\mathbf{W}'_s \in \mathbb{R}^{(C/N_h) \times C}$ denotes learnable weights and $\mathbf{A}_{hs}$ denotes the attention weights which are normalized as $\sum_s \mathbf{A}_{h,s} = 1$. The resulting point embedding $\mathbf{z}$ serves as input to both rendering heads and detection heads described in the following sections.

## 3.2 POINT-BASED 3D SCENE RECONSTRUCTION & PERCEPTION

**Volumetric Rendering** To support 3D scene understanding for downstream tasks in autonomous driving, we first optimize radiance fields in a self-supervised manner (Yang et al., 2024), using the signed distance function (SDF) and RGB reconstruction to represent 3D geometry and appearance. Given a set of sampled points along each ray and its embedded features $\{\mathbf{z}_{ij}\}$, RGB color values of 3D points $\mathbf{x}_j$ are predicted by $c_j = \phi_{rgb}(\mathbf{z}_j, \mathbf{d}_i)$, and its signed distance $s_j$ extracted by signed distance function $\phi_{sdf}(\mathbf{z}_j)$ is transformed into opacity $\alpha_j$ derived with:

$$\alpha_j = \max\left(\frac{\Phi_\omega(\phi_{sdf}(\mathbf{z_j})) - \Phi_\omega(\phi_{sdf}(\mathbf{z}_{j+1}))}{\Phi_\omega(\phi_{sdf}(\mathbf{z}_j))}, 0\right), \tag{3}$$

where $\Phi_\omega(x) = (1 + e^{-\omega x})^{-1}$ is the sigmoid function with a learnable parameter $\omega$. Then, the unbiased and occlusion-aware weights (Wang et al., 2021) $w_j = T_j \alpha_j$ is computed from $\alpha_j$, where $T_j = \prod_{k=1}^{j-1} (1 - \alpha_k)$ is the accumulated transmittance. The final color and depth values are computed by accumulating the contributions of 3D points sampled along ray $\mathbf{r}_i$, weighted by the probability distribution $\{w_j\}$:

$$\hat{\mathbf{C}}(\mathbf{r}_i) = \sum_{j=1}^D w_j \mathbf{c}_j, \quad \hat{D}(\mathbf{r}_i) = \sum_{j=1}^D w_j t_j, \tag{4}$$

where $\hat{\mathbf{C}}(\mathbf{r}_i)$ and $\hat{D}(\mathbf{r}_i)$ denote the predicted color and depth corresponding to the ray $\mathbf{r}_i$, respectively.

To optimize the neural radiance field, we employ a combination of RGB reconstruction, depth supervision, and multi-view consistency losses. We adopt the standard volumetric rendering loss from NeRFs, comparing the rendered color $\hat{\mathbf{C}}(\mathbf{r}_i)$ against the ground truth pixel color $\mathbf{C}(\mathbf{r}_i)$ for sampled rays $\mathcal{R} = \{\mathbf{r}_i\}$. To further constrain the 3D geometry, we leverage explicit depth supervision (Deng et al., 2022; Yang et al., 2024) for $\mathbf{r}_i$ against LiDAR measurements $D_{lidar}(\mathbf{r}_i)$ where available. Furthermore, while LiDAR provides direct supervision, it suffers from sparse scan patterns and cannot capture regions such as the sky, transparent surfaces (*e.g.*, windows), or distant backgrounds where depth is undefined or unprojectable. To address this without additional annotations (Yang et al., 2023b) or distillation from 2D foundation models (Oquab et al., 2023; Kirillov et al., 2023), we

further enforce multi-view consistency (Godard et al., 2019; Cao & De Charette, 2023) by minimizing the discrepancy in predicted depth distributions across different views as:

$$\mathcal{L}_{reproj} = \frac{1}{|\mathcal{R}|} \sum_{\mathbf{r}_i \in \mathcal{R}} \sum_{\mathbf{x}_j \in \mathbf{r}_i} w_j |I_t(\mathbf{r}_i) - I_s(\pi_s(\mathbf{x}_j))|, \tag{5}$$

where $I_t(\mathbf{r}_i)$ denotes the color value of a pixel in a target or current image $I_t$ corresponding to the ray $\mathbf{r}_i$. $\pi_s(\mathbf{x})$ denotes the projection matrix from 3D points to 2D pixels on a source image $I_s$, such as a previous $I_{t-1}$ or future image $I_{t+1}$. Consequently, the sampled 3D point $\mathbf{x}_j = \mathbf{o}_i + t_j \mathbf{d}_i$ along the ray $\mathbf{r}_i$ is projected on the source image, and the corresponding pixel color $I_s(\pi_s(\mathbf{x}_j))$ is compared with $I_t(\mathbf{r}_i)$ in weighted sum $\{w_j\}$. The overall loss for pre-training consists of RGB reconstruction loss, depth supervision loss, and reprojection loss:

$$\mathcal{L}_{pretrain} = \lambda_{rgb} \mathcal{L}_{rgb} + \lambda_{depth} \mathcal{L}_{depth} + \lambda_{reproj} \mathcal{L}_{reproj} \tag{6}$$

where $\lambda_{rgb}$, $\lambda_{depth}$, and $\lambda_{reproj}$ are the loss scale factors for each pre-training loss. $\mathcal{L}_{rgb}$ is RGB reconstruction loss and $\mathcal{L}_{depth}$ is depth estimation loss directly supervised by LiDAR measurements.

**Open-World Perception** Unlike view-dependent volumetric rendering, perception tasks require comprehensive spatial coverage of the vehicle's surroundings. All we need to do with NeRP3D is scatter the points $\{\mathbf{x}\} \in \mathbb{R}^{N \times 3}$ throughout the space and reshape the resulting representations $\{\mathbf{z}\} \in \mathbb{R}^{N \times C}$ from Eq. 2 to be compatible with task-specific detection heads, for example, $\{\mathbf{z}\} \in \mathbb{R}^{(X \times Y \times Z) \times C}$ for occupancy prediction. This straightforward adaptation maintains the enhanced geometric and appearance information learned during pre-training while enabling seamless integration with established perception architectures.

## 4 EXPERIMENTS

We demonstrate NeRP3D on the nuScenese (Caesar et al., 2020) dataset against the *state-of-the-art* NeRF-based pre-training approaches as well as comparable methods. Our experiments are designed to assess both pre-trained 3D representations by scene reconstruction and the effectiveness of finetuning for downstream tasks.

### 4.1 DATASET

We conduct experiments using the nuScenes dataset (Caesar et al., 2020), which provides 700, 150, and 150 scenes for training, validation, and testing, respectively. We follow this data split for both the pretext and downstream tasks. Each scene provides 6 RGB camera images that cover a full 360° field of view, along with a 32-beam LiDAR point cloud and 3D radar point cloud data. The key samples are annotated at 2 Hz and support multiple tasks for autonomous driving, including 3D object detection, HD map construction, and segmentation. Recently, the annotations for occupancy prediction have been made available through Occ3D (Tian et al., 2023) and SurroundOcc (Wei et al., 2023b), providing dense 3D semantic occupancy labels. In our experiments, we adopt the Occ3D benchmark for the occupancy prediction.

Moreover, to evaluate generalization across different data distributions and sensor configurations, we additionally utilize Argoverse 2 (AV2) (Wilson et al., 2023) dataset. AV2 provides 1,000 driving sequences with a distinct sensor suite comprising seven high-resolution ring cameras ($2048 \times 1550$) covering a 360° field of view and two 32-beam LiDARs. This setup introduces significant domain shifts in environmental statistics and sensor layouts compared to nuScenes (Caesar et al., 2020). This distinct setup serves to assess the model's robustness to domain changes and its data efficiency under limited supervision. For our experiments, we resized the input images to $800 \times 450$ and utilized only a $1/4$ subset of the training data.

### 4.2 EVALUATION METRICS

We evaluate performance across two pretext scene reconstruction tasks and three downstream 3D perception tasks by following standard evaluation protocols for each task to ensure comparability with existing methods.

Table 1: **Downstream detection performance**

(a) **3D object detection**

| Method | Pre-train | NDS↑ | mAP↑ |
|---|---|---|---|
| UVTR-C | ImageNet | 44.1 | 37.2 |
| BEVFormerV2 | ImageNet | 46.7 | 39.6 |
| TPVFormer† | SelfOcc | 33.5 | 31.0 |
| UVTR-C† | UniPAD | 37.1 | 33.7 |
| **NeRP3D†** | Ours | **39.2** | **35.8** |
| UVTR-C | UniPAD | 45.5 | 41.6 |
| **NeRP3D** | Ours | **47.3** | **42.8** |

(b) **Occ prediction**

| Method | mIoU |
|---|---|
| BEVDet | 19.38 |
| BEVFormer | 26.88 |
| TPVFormer | 27.83 |
| CTF-Occ | 28.53 |
| SelfOcc | 29.65 |
| UniPAD | 34.05 |
| **NeRP3D** | **35.49** |

(c) **HD map construction**

| Method | Pre-train | Epochs | mAP |
|---|---|---|---|
| HDMapNet | ImageNet | 30 | 23.0 |
| VectorMapNet | ImageNet | 110 | 40.9 |
| MapTR-tiny | ImageNet | 24 | 49.9 |
| TPVFormer | SelfOcc | 24 | 53.9 |
| UVTR-C | UniPAD | 24 | 57.8 |
| **NeRP3D** | Ours | 24 | **59.1** |

**Scene Reconstruction Tasks**   We compare scene reconstruction quality by generating rendered RGB and depth images 1:4 the size of the input images. RGB reconstructed images are evaluated for all rendered pixels by Peak Signal-to-Noise Ratio (PSNR), Structural Similarity Index Measure (SSIM), and Learned Perceptual Image Patch Similarity (LPIPS), following standard NeRF evaluation protocols. For depth estimation, we report relative errors (AbsRel & SqRel), root mean squared error (RMSE & RMSE log), and accuracy under threshold $\delta$ metrics up to $80m$, only for pixels where the lidar point cloud with 20 sweeps is projected.

**Downstream Tasks**   We evaluate the performance of 3D object detection using the mean Average Precision (mAP) and nuScenes Detection Score (NDS) under the standard nuScenes evaluation protocol. The perception range for object detection is set to $[-51.2m, 51.2m]$ along both the X and Y axes. For vectorized HD map construction, we follow the nuScenes map annotation benchmark and report mAP under *Chamfer* distance thresholds ($\tau \in \{0.5, 1.0, 1.5\}$). The evaluation range is set to $[-15.0m, 15.0m]$ for the X axis and $[-30.0m, 30.0m]$ for the Y axis. Occupancy prediction aims to predict the semantic classes of $0.4m \times 0.4m \times 0.4m$ voxels covering $[-40m, 40m]$ in both the X and Y axes and $[-1.0m, 5.4m]$ along the Z axis. The prediction result is evaluated using mean Intersection over Union (mIoU) across 17 semantic classes.

## 4.3   IMPLEMENTATION DETAILS

To ensure fair comparison with prior works (Yang et al., 2024; Huang et al., 2024), we adopt identical pre-training architectures and detection heads. We leverage NeuS (Wang et al., 2021) for radiance field pre-training, following previous studies. Furthermore, we conduct downstream tasks based on UVTR-C (Li et al., 2022), MapTR (Liao et al., 2023), and Occ3D (CTF-Occ) (Tian et al., 2023) for 3D object detection, HD map construction, and occupancy prediction, respectively. Class-balanced sampling (CBGS) or specialized data augmentations are not applied for finetuning, and all downstream tasks are trained and evaluated using single-timestep images only, without temporal information or frame stacking.

Our implementation builds upon the MMDetection3D (Contributors, 2020) framework, and training is conducted on 4 NVIDIA A6000 GPUs. The input image resolution varies by tasks, set to $1600 \times 900$ for object detection and $800 \times 450$ for rendering, HD map construction, and occupancy prediction. We both pre-train and fine-tune the model for 24 epochs using the AdamW optimizer, with an initial learning rate of 2e-4 and a weight decay of 0.01. The loss scale factors are set to $\lambda_{rgb} = \lambda_{depth} = \lambda_{reproj} = 10$. Unless otherwise specified, we fine-tune the models on a 1/2 subset for 12 epochs with $800 \times 450$ images in ablation studies.

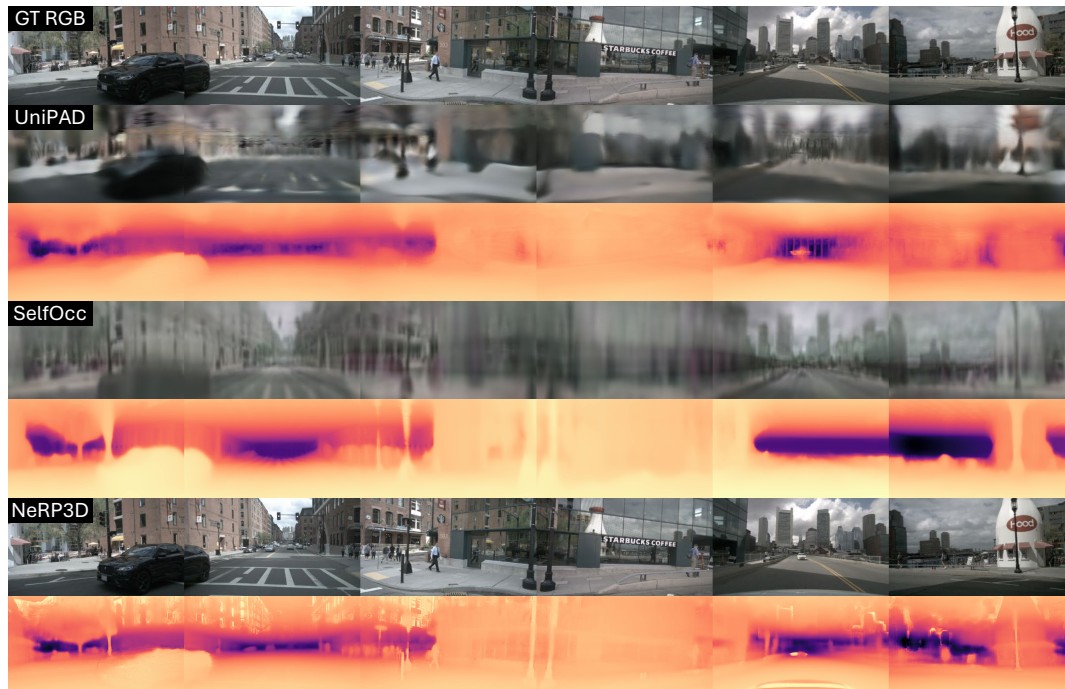

Figure 4: Qualitative comparison on rendered RGB & depth. NeRP3D outperforms *state-of-the-art* methods on both RGB and depth reconstruction. Our approach maintains high fidelity in urban scenes without any blur and pattern artifacts. For depth estimation, NeRP3D distinguishes individual people in crowded areas rather than merging them into indistinct blobs, and precisely captures thin structures such as poles that are often missed or reconstructed as thick structures by competing methods.

Table 2: **Pretext scene reconstruction performance**

<table>
<tr><td colspan="5" align="center">(a) Depth estimation</td></tr>
<tr><td>Method</td><td>Abs Rel↓</td><td>Sq Rel↓</td><td>RMSE↓</td><td>RMSE log↓</td></tr>
<tr><td>SelfOcc</td><td>0.311</td><td>3.808</td><td>8.503</td><td>0.391</td></tr>
<tr><td>SelfOcc*</td><td>0.215</td><td>2.743</td><td>6.706</td><td>0.316</td></tr>
<tr><td>UniPAD</td><td>0.218</td><td>2.512</td><td>7.937</td><td>0.356</td></tr>
<tr><td>NeRP3D</td><td>0.183</td><td>2.274</td><td>7.884</td><td>0.353</td></tr>
</table>

<table>
<tr><td colspan="4" align="center">(b) RGB reconstruction</td></tr>
<tr><td>Method</td><td>PSNR↑</td><td>SSIM↑</td><td>LPIPS↓</td></tr>
<tr><td>SelfOcc</td><td>18.82</td><td>0.536</td><td>0.657</td></tr>
<tr><td>UniPAD</td><td>21.14</td><td>0.549</td><td>0.634</td></tr>
<tr><td>NeRP3D</td><td>33.42</td><td>0.969</td><td>0.070</td></tr>
</table>

## 4.4 MAIN RESULTS

**3D Object Detection**   We compare NeRP3D with previous 3D object detection approaches (Li et al., 2024; 2022; Liu et al., 2022a; Shu et al., 2023; Yang et al., 2023a; Yan et al., 2023) on the nuScenes *val* set. To compare with previous NeRF-based pre-training methods on detection, we follow the fine-tuning framework of UniPAD (Yang et al., 2024) and also reproduce the results of both UVTR-C (UniPAD) (Li et al., 2022; Yang et al., 2024) and TPVFormer (SelfOcc) (Huang et al., 2023; 2024) by replacing the NeRF network for pre-training with UVTR's object detection head. † in Tab .1 (a) denotes the result evaluated on input resolutions of $800 \times 450$. Compared to the *state-of-the-art* NeRF-based self-supervision methods, our method outperforms 1.8 mAP and 2.1 NDS on $800 \times 450$ 1.2 mAP and 1.8 NDS on $1600 \times 900$ over UniPAD, as shown in Tab. 1 (a). This improvement stems from NeRP3D's ability to learn fine-grained 3D representations, which enables more precise localization of bounding boxes and better separation of nearby objects, as qualitatively suggested by the detailed features in Fig. 1 and sharp reconstructions in Fig. 4.

**Occupancy Prediction**   In Tab. 1 (b), we evaluate the performance of our method on Occ3D-nuScenes for 3D occupancy prediction. Similar to 3D object detection, we fine-tune the backbones with the same occupancy prediction head (Tian et al., 2023) after pre-training. Our approach inherits NeRF's strength in modeling fine-grained representations, leading to improved mIoU and consistent

Table 3: **Zero-shot scene reconstruction performance (Argoverse 2 → nuScenes)**

| Method | Abs Rel↓ | Sq Rel↓ | RMSE↓ | RMSE log↓ | PSNR↑ | SSIM↑ | LPIPS↓ |
|---|---|---|---|---|---|---|---|
| UniPAD | 0.985 | 11.767 | 14.963 | 4.390 | 18.668 | 0.432 | 0.577 |
| **NeRP3D** | **0.626** | **6.251** | **10.728** | **0.921** | **28.238** | **0.905** | **0.111** |

gains over UniPAD and SelfOcc. As a result, our NeRP3D outperforms UniPAD and SelfOcc by 2.8 and 9.2 mIoUs, respectively. The continuous and high-fidelity representations learned by NeRP3D are particularly beneficial for this dense prediction task, enabling the model to accurately discern object boundaries and capture intricate geometric details often missed by other methods.

**HD Map Reconstruction**   We evaluate the accuracy of HD map construction to assess each method's capability for understanding static driving environments, particularly in detecting road boundaries, dividers, and pedestrian crossings. To facilitate this task, we commonly utilized the detection head of MapTR (Liao et al., 2023) for fair comparison. As shown in Tab. 1 (c), our method achieves improved mAP compared to both UniPAD and SelfOcc, with gains of 1.3 and 5.2 mAP, respectively. HD map reconstruction is particularly challenging as it requires a nuanced semantic understanding to differentiate map elements like pedestrian crossings that are geometrically coplanar with the drivable surface. As visually evidenced in Fig. 1, the feature representations from NeRP3D make these elements distinctly separable, which is critical for precise map construction.

**RGB & Depth Reconstruction**   To validate the effectiveness of the pre-training, the performance of NeRP3D on the pretext tasks is also compared with the previous NeRF-based pre-training methods (Yang et al., 2024; Huang et al., 2024) on the nuScenes *val* set. As shown in Tab. 2, NeRP3D achieves remarkable enhancements in both depth estimation and RGB reconstruction. More specifically, the qualitative depth maps in Fig. 4 consistently demonstrate that our method yields more accurate and detailed depth estimations, particularly in complex regions, whereas UniPAD and Self-Occ struggle to resolve fine structures and depth discontinuities. For RGB reconstruction, UniPAD generates blurry and imprecise reconstructions lacking detailed textures, while SelfOcc produces grayish images with unidentified vertical patterns. In contrast, our approach reconstructs sharper images with rich colors, closely matching the ground truth without introducing patterned signals.

**Generalization**   To assess the robustness of our method against domain shifts and varying sensor configurations, we conducted cross-dataset transfer experiments using Argoverse 2 (AV2) (Wilson et al., 2023) for pre-training and nuScenes for evaluation. AV2 possesses distinct camera geometries and environmental statistics compared to nuScenes, serving as a rigorous testbed for generalization.

We first evaluated zero-shot scene reconstruction by directly applying the AV2 (Wilson et al., 2023) pre-trained weights to nuScenes (Caesar et al., 2020) without any fine-tuning. As presented in Tab. 3, NeRP3D demonstrates remarkable robustness, achieving an Abs Rel of 0.626 and PSNR of 28.24, significantly outperforming UniPAD (Abs Rel 0.985, PSNR 18.67). While the view transformation method (Yang et al., 2024) suffers from severe degradation due to its dependency on fixed grid priors aligned with specific sensor layouts, NeRP3D's continuous point-based architecture effectively adapts to new sensor geometries. Qualitative results in Fig. 10 of Appendix further visualize this, showing that NeRP3D preserves structural details while the view transformation method produces blurry artifacts.

Moreover, we evaluated the transferability for 3D object detection. When pre-trained on AV2 (Wilson et al., 2023) and fine-tuned on nuScenes (Caesar et al., 2020), NeRP3D achieved 27.46 mAP, surpassing UniPAD (Yang et al., 2024) (26.29 mAP) by a significant margin. This confirms that NeRP3D learns universal geometric representations that are not overfitted to specific sensor configurations or dataset distributions but are effectively transferable across domains.

Overall, these results demonstrate that our approach effectively leverages the inherent advantages of continuous and fine-grained representations derived from NeRF. NeRP3D not only significantly benefits pretext scene reconstruction tasks and downstream detection tasks but also ensures robust generalization across different sensor configurations and data distributions for autonomous driving. More comprehensive comparison and quantitative analysis of the experimental results are provided by Tab. 4−8 in Appendix A and B.

### 4.5 Ablation Studies

We conduct comprehensive ablation studies to analyze different model variants and evaluate their impact. Ablation results are reported in Appendix C and summarized in the following sections.

**Cross-Task Generalization**   We further investigate whether the learned 3D representation remains valid across different task objectives. By performing volumetric rendering using the backbone fine-tuned for occupancy prediction, we observe that NeRP3D successfully retains structural details, whereas view-transformation methods suffer from catastrophic forgetting, collapsing into mean regression. This confirms that NeRP3D learns a task-agnostic continuous representation that preserves geometric fidelity regardless of downstream optimization pressure.

**Adaptability**   View transformation is dependent on the range and voxel size, leading to severe performance degradation if the voxel-related parameters are changed against pre-training. In contrast, NeRP3D aims for a continuous representation without voxel-related parameters, and variations only correspond to simple changes in the range of interest.

**Effectivness**   We analyze the effectiveness of NeRP3D in reducing the reliance on annotations by comparing previous works, ranging from the full dataset to a 1/8 subset. Consequently, NeRP3D maintains strong detection performance even with significantly reduced supervision, indicating the robustness of its NeRF-based pre-training.

**Multi-view Consistency**   LiDAR-based supervision ensures more consistent depth estimation accuracy. However, we found that the sparsity and scan patterns of LiDAR are ultimately insufficient for reconstructing dense 3D geometry. To address LiDAR's sparsity and patterns, we not only rely on LiDAR supervision but also consider multi-view consistency and our sampling strategy tailored to this approach.

**Design Validation**   We verify the necessity of our architectural choices through comprehensive comparisons. First, applying NeRF pre-training to existing point-based detectors (Liu et al., 2023a) fails to transfer knowledge due to query mismatch, confirming the importance of our unified design. Second, comparisons between SDF (Wang et al., 2021) and density (Mildenhall et al., 2021) priors validate that SDF enforces clearer object boundaries beneficial for perception. Finally, we demonstrate that deformable attention outperforms standard attention by providing a necessary locality inductive bias, ensuring that the points remain faithful to their local spatial context.

## 5 Conclusion

In this paper, we present NeRP3D, a novel point-based 3D architecture for scene reconstruction and downstream perception tasks for autonomous driving. Our approach addresses the fundamental misalignment between view transformation and neural radiance fields. Through its NeRF-resembled design, NeRP3D fully inherits NeRF's continuous representation capabilities, enabling the model to maintain consistent geometric and appearance information at arbitrary spatial locations across both scene reconstruction and open-world perceptions. Although NeRP3D outperforms previous approaches, it struggles with depth beyond its ROI, relying on LiDAR. Additionally, its point-based architecture incurs high computational costs from adapting NeRF's output to existing detection heads. Future enhancements include temporal RGB reconstruction for consistency, density/opacity filtering for efficiency, and Gaussian splatting for real-time performance with point queries.

## 6 Acknowledgement

This work was supported by the National Research Foundation of Korea(NRF) grant funded by the Korea government (MSIT) and the Ministry of Trade, Industry and Resources (MOTIR) of the Republic of Korea. (2022R1A2C200494414, RS-2025-25448249)

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
