Table 4: **3D object detection** on the nuScenes *val* set. † denotes the result evaluated on input resolutions of $800 \times 450$ using MMDetection3D (Contributors, 2020) by integrating UVTR detection head (Li et al., 2022; Yang et al., 2024). The other results are based on $1600 \times 900$ input resolution.

| Method | Pre-train | NDS↑ | mAP↑ | mATE↓ | mASE↓ | mAOE↓ | mAVE↓ | mAAE↓ |
|---|---|---|---|---|---|---|---|---|
| BEVFormer-S | ImageNet | 44.8 | 37.5 | - | - | - | - | - |
| UVTR-C | ImageNet | 44.1 | 37.2 | 0.735 | 0.269 | 0.397 | 0.761 | 0.193 |
| PETR | ImageNet | 44.2 | 37.0 | 0.711 | 2.670 | 0.383 | 0.865 | 0.201 |
| 3DPPE | ImageNet | 45.8 | 39.1 | - | - | - | - | - |
| BEVFormerV2 | ImageNet | 46.7 | 39.6 | 0.709 | 0.274 | 0.368 | 0.768 | 0.196 |
| CMT-C | ImageNet | 46.0 | 40.6 | - | - | - | - | - |
| TPVFormer† | SelfOcc | 33.5 | 31.0 | 0.785 | 0.285 | 0.729 | 1.232 | 0.399 |
| UVTR-C† | UniPAD | 37.1 | 33.7 | 0.734 | 0.283 | 0.603 | 1.250 | 0.359 |
| **NeRP3D†** | Ours | **39.2** | **35.8** | 0.719 | 0.288 | 0.640 | 0.977 | 0.250 |
| UVTR-C | UniPAD | 45.5 | 41.6 | 0.674 | 0.277 | 0.418 | 0.930 | 0.234 |
| UVTR-C | GaussianPretrain | 47.2 | 41.7 | 0.676 | 0.278 | 0.394 | 0.815 | 0.200 |
| **NeRP3D** | Ours | **47.3** | **42.8** | 0.664 | 0.276 | 0.425 | 0.811 | 0.196 |

Table 5: **3D occupancy prediction**. We compare our method against *state-of-the-art* occupancy prediction approaches on the Occ3d-nuScenes *val* set. Results for BEVDet, BEVFormer, TPVFormer, and CTF-Occ are directly taken from Occ3d (Tian et al., 2023). † denotes the result reproduced using MMDetection3D (Contributors, 2020) on input resolutions of $800 \times 450$. $*$ denotes that the result is directly taken from VisionPAD (Zhang et al., 2025), which is pre-trained only with camera modality and evaluated on input resolutions of $1600 \times 900$.

| Method | mIoU | car | bus | bicycle | barrier | vegetation | const. veh. | motorcycle | pedestrian | traffic cone | trailer | truck | drive. suf. | other flat | sidewalk | terrain | manmade | others |
|---|---|---|---|---|---|---|---|---|---|---|---|---|---|---|---|---|---|---|
| BEVDet | 19.4 | 34.5 | 32.3 | 0.2 | 30.3 | 15.1 | 13.0 | 10.3 | 10.4 | 6.3 | 8.9 | 23.7 | 52.7 | 24.6 | 26.1 | 22.3 | 15.0 | 4.4 |
| BEVFormer | 26.9 | 42.4 | 40.4 | 17.9 | 37.8 | 17.7 | 7.4 | 23.9 | 21.8 | 21.0 | 22.4 | 30.7 | 55.4 | 28.4 | 36.0 | 28.1 | 20.0 | 5.9 |
| TPVFormer | 27.8 | 45.9 | 40.8 | 13.7 | 38.9 | 16.8 | 17.2 | 20.0 | 18.9 | 14.3 | 26.7 | 34.2 | 55.7 | 35.5 | 37.6 | 30.7 | 19.4 | 7.2 |
| CTF-Occ | 28.5 | 42.2 | 38.3 | 20.6 | 39.3 | 18.0 | 16.9 | 24.5 | 22.7 | 21.1 | 23.0 | 31.1 | 53.3 | 33.8 | 38.0 | 33.2 | 20.8 | 8.1 |
| SelfOcc† | 29.7 | 43.8 | 40.0 | 10.0 | 36.3 | 30.6 | 13.7 | 11.8 | 16.5 | 15.7 | 23.2 | 29.3 | 79.1 | 37.3 | 47.7 | 28.0 | 34.8 | 6.2 |
| UniPAD† | 34.1 | 45.8 | 42.3 | 13.0 | 39.7 | 38.1 | 19.4 | 14.3 | 20.0 | 17.7 | 27.4 | 33.1 | 80.0 | 38.7 | 49.4 | 50.6 | 42.8 | 6.5 |
| VisionPAD* | 35.4 | - | - | - | - | - | - | - | - | - | - | - | - | - | - | - | - | - |
| **NeRP3D†** | **35.5** | 49.4 | 43.9 | 15.0 | 41.0 | 38.8 | 19.2 | 20.0 | 23.6 | 16.5 | 27.9 | 36.7 | 81.0 | 37.4 | 49.8 | 53.6 | 43.9 | 5.5 |

# A DOWNSTREAM DETECTION TASKS

A detailed analysis of NeRP3D's performance is provided on three downstream perception tasks: 3D object detection, 3D occupancy prediction, and HD map construction. We expand upon the results presented in Sec. 4.4 and Tab. 1, with a focus on comprehensive comparisons against state-of-the-art methods, including those leveraging 3DGS (3D Gaussian Splatting)-based pre-training.

As shown in Tab. 4, NeRP3D achieves state-of-the-art performance in 3D object detection among NeRF-based pre-training methods, with an NDS of 47.3 and an mAP of 42.8. This represents a significant improvement over UniPAD, with gains of 1.8 NDS and 1.2 mAP when both are fine-tuned on the UVTR-C detector. Crucially, NeRP3D also outperforms GaussianPretrain (Xu et al., 2024), which still relies on a view transformation backbone. In comparison, NeRP3D achieves a higher NDS (47.3 vs. 47.2) and a more substantial lead in mAP (42.8 vs. 41.7). The enhanced performance is attributed to NeRP3D's fine-grained 3D representation, which provides the necessary detail to identify far or occluded targets and resolve individuals within dense crowds, as shown in Fig. 8

For 3D occupancy prediction, NeRP3D's ability to model continuous geometry and appearance translates into superior performance. As demonstrated in Tab. 5, our method achieves an mIoU of 35.5, surpassing both UniPAD (34.1 mIoU) and SelfOcc (29.7 mIoU) by a significant margin. We further compare NeRP3D with VisionPAD (Zhang et al., 2025), a vision-centric pre-training also based on 3D Gaussians. Even though VisionPAD is pre-trained only with camera modality, but

Table 6: **HD map construction** on the nuScenes *val* set. "C" and "L" denote camera and Li-DAR modalities, respectively. Results for HDMapNet and VectorMapNet are directly taken from MapTR(Liao et al., 2023).

| Method | Modality | Pre-train | Epochs | mAP | $AP_{ped}$ | $AP_{divider}$ | $AP_{boundary}$ |
|---|---|---|---|---|---|---|---|
| HDMapNet | C | ImageNet | 30 | 23.0 | 14.4 | 21.7 | 33.0 |
| HDMapNet | L | ImageNet | 30 | 24.1 | 10.4 | 24.1 | 37.9 |
| HDMapNet | C & L | ImageNet | 30 | 31.0 | 16.3 | 29.6 | 46.7 |
| VectorMapNet | C | ImageNet | 110 | 40.9 | 36.1 | 47.3 | 39.3 |
| VectorMapNet | L | ImageNet | 110 | 34.0 | 25.7 | 37.6 | 38.6 |
| VectorMapNet | C & L | ImageNet | 110 | 45.2 | 37.6 | 50.5 | 47.5 |
| MapTR-tiny | C | ImageNet | 24 | 49.9 | 52.0 | 45.3 | 52.4 |
| TPVFormer | C | SelfOcc | 24 | 53.9 | 47.8 | 55.6 | 58.3 |
| UVTR-C | C | UniPAD | 24 | 57.8 | **54.8** | 58.5 | 61.5 |
| **NeRP3D** | C | Ours | 24 | **59.1** | 52.9 | **62.2** | **62.2** |

Table 7: **Depth estimation** on nuScenes *val* set. We conduct evaluation at a downsampled resolution of $114 \times 64$ for EmerNeRF (Yang et al., 2023b) and DistillNeRF (Wang et al., 2024) and $400 \times 224$ for others. † denotes per-scene optimization, not feedforward model. * denotes only *depth-optimized* variant of SelfOcc (Huang et al., 2024). The results of EmerNeRF and DistillNeRF are taken from the paper of DistillNeRF.

| Method | Abs Rel↓ | Sq Rel↓ | RMSE↓ | RMSE log ↓ | $\delta < 1.25$ ↑ | $\delta < 1.25^2$ ↑ | $\delta < 1.25^3$ ↑ |
|---|---|---|---|---|---|---|---|
| EmerNeRF† | 0.073 | 0.346 | 2.696 | 0.159 | 0.942 | 0.975 | 0.986 |
| DistillNeRF | 0.248 | 3.090 | 6.096 | 0.312 | 0.704 | 0.885 | 0.947 |
| DistillNeRF-D | 0.233 | 2.890 | 5.890 | 0.296 | 0.703 | 0.881 | 0.945 |
| DistillNeRF-DV | 0.223 | **1.776** | **5.461** | **0.293** | 0.763 | **0.903** | **0.961** |
| SelfOcc | 0.311 | 3.808 | 8.503 | 0.391 | 0.641 | 0.803 | 0.888 |
| SelfOcc* | 0.215 | 2.743 | 6.706 | 0.316 | 0.753 | 0.875 | 0.932 |
| UniPAD | 0.218 | 2.512 | 7.937 | 0.356 | 0.763 | 0.869 | 0.921 |
| NeRP3D | **0.183** | 2.274 | 7.884 | 0.353 | **0.799** | 0.883 | 0.926 |

evaluated on the higher resolution $1600 \times 900$, NeRP3D achieves a competitive overall mIoU (35.5 vs. 35.4). A class-level breakdown reveals that NeRP3D shows notable improvements in thin and small categories, as shown in Fig. 9, such as bicycle (15.0 vs. 13.0), motorcycle (20.0 vs. 14.3), and pedestrian (23.6 vs. 20.0).

The comprehensive results for downstream perception tasks indicate that our NeRP3D, which avoids the conflicting priors between the pre-training method and 3D backbone, enables the learning of continuous and fine-grained 3D representations that directly benefit downstream detection tasks.

# B   PRETEXT SCENE RECONSTRUCTION TASKS

The overall performance of RGB reconstruction and depth estimation is compared with previous NeRF-based pre-training methods (Yang et al., 2024; Huang et al., 2024) and comparable methods on the nuScenes *val* set, as shown in Tab. 7 and 8. Specifically, EmerNeRF (Yang et al., 2023b) is a *per-scene* optimization model, and the variants of DistillNeRF (Wang et al., 2024) are *without* distillation, *with* depth distillation (noted as "D"), and *with* virtual camera distillation (noted as "V").

The depth estimation results in Tab. 7 demonstrate clear benefits from our NeRF-inherited representation learning. SelfOcc* shows competitive depth estimation, but this variant does not support

| Method | PSNR ↑ | SSIM ↑ | LPIPS ↓ |
|---|---|---|---|
| EmerNeRF | 30.88 | 0.879 | - |
| DistillNeRF-D | 30.11 | 0.917 | - |
| SelfOcc | 18.82 | 0.536 | 0.657 |
| UniPAD | 21.14 | 0.549 | 0.634 |
| **NeRP3D** | **33.42** | **0.969** | **0.070** |

Table 8: **RGB reconstruction** on nuScenes *val* set at a resolution of $228 \times 114$ for EmerN-eRF (Yang et al., 2023b) and DistillNeRF (Wang et al., 2024) and $400 \times 225$ for others. The results of EmerNeRF and DistillNeRF are taken from the paper of DistillNeRF.

Table 9: **Multi-resolution reconstruction analysis.** We evaluate reconstruction quality across varying image scales (from $1/16$ to $1/4$) to isolate the impact of discretization levels on representational fidelity.

| Method | Scale | PSNR↑ | SSIM↑ | LPIPS↓ |
|---|---|---|---|---|
| UniPAD (Yang et al., 2024) | 1/16 | 23.55 | 0.796 | 0.250 |
| | 1/8 | 22.49 | 0.664 | 0.442 |
| | 1/4 | 21.14 | 0.549 | 0.634 |
| **NeRP3D** | 1/16 | 26.00 | 0.862 | 0.116 |
| | 1/8 | 29.40 | 0.919 | 0.090 |
| | 1/4 | 33.42 | 0.969 | 0.070 |

Table 10: **Evaluation of Cross-Task Generalization and Structural Retention.** (a) Per-pixel evaluation: Standard metrics measuring absolute reconstruction errors, which are sensitive to scale shifts. (b) Structural evaluation: Scale-invariant and perceptual metrics assessing geometric fidelity and structural integrity, independent of feature scale variations induced during fine-tuning.

(a) **Per-pixel evaluation**

| Method | Abs Rel↓ | Sq Rel↓ | RMSE↓ | RMSE log↓ | PSNR↑ | SSIM↑ |
|---|---|---|---|---|---|---|
| UniPAD (Yang et al., 2024) | 0.477 | 6.914 | 15.104 | 1.056 | 11.623 | 0.283 |
| **NeRP3D** | 2.192 | 12.372 | 19.459 | 1.185 | 9.308 | 0.135 |

(b) **Structural and scale-invariant evaluation**

| Method | SI RMSE↓ | Grad Loss↓ | GMSD↓ | LPIPS↓ | PSNR-HM↑ | SSIM-HM↑ |
|---|---|---|---|---|---|---|
| UniPAD (Yang et al., 2024) | 0.859 | 90.164 | 0.306 | 0.863 | 12.319 | **0.300** |
| **NeRP3D** | **0.643** | **83.739** | **0.289** | **0.671** | **12.839** | 0.285 |

RGB reconstruction. On the other hand, the variant of SelfOcc that supports both RGB and depth reconstruction exhibits comparatively lower accuracy. Compared to UniPAD, our method achieves better performance across multiple metrics, such as AbsRel (0.183 vs. 0.218), SqRel (2.274 vs. 2.512), and RMSE (7.884 vs. 7.937). Moreover, accuracy within specific depth thresholds ($\delta$ metrics) further underscores the robustness of our model in reconstructing precise depth values. When compared with DistillNeRF, which is specifically designed for scene reconstruction, our NeRP3D achieves competitive depth estimation accuracy despite not relying on dense depth maps obtained from per-scene optimization (Yang et al., 2023b) or distillation from 2D foundation models (Radford et al., 2021; Oquab et al., 2023).

For RGB reconstruction, NeRP3D significantly outperforms previous approaches, as shown in Tab. 8. Compared to previous feedforward methods and EmerNeRF, PSNR and SSIM are improved by 33.42 and 0.969, respectively. Our method also notably reduces LPIPS, reflecting more perceptually accurate reconstructions over UniPAD and SelfOcc by 0.070.

To quantitatively verify the conflicting prior between discrete view transformation and continuous neural rendering representations, we evaluated reconstruction performance across varying resolutions as shown in Tab. 9. The view transformation method (UniPAD (Yang et al., 2024)) degrades at higher resolutions, confirming that discrete voxel grids act as a "low-pass filter". As a result, UniPAD masks errors at coarse scales but fails to capture high-frequency details due to the rigid bottleneck. In contrast, NeRP3D demonstrates superior representational fidelity with a widening performance gap at finer scales. This quantitatively proves that our continuous architecture resolves the structural mismatch, successfully modeling fine-grained geometry that fixed grids cannot capture.

## C ABLATION STUDIES

### C.1 CROSS TASK GENERALIZATION

We investigate whether the learned 3D representations remain valid across different task objectives, specifically evaluating the "Radiance Modeling Ability" of the backbone after fine-tuning for occupancy prediction. In this experiment, we utilize the backbone encoder fine-tuned for the downstream task while keeping the pre-trained rendering heads (RGB and SDF decoder) frozen.

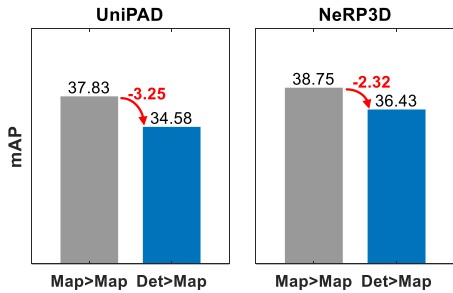

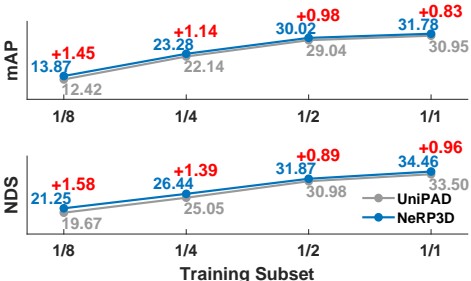

Figure 5: Comparison of performance variation with changes in detection range between the pre-training and fine-tuning phases.

Figure 6: Comparison of pre-training effectiveness: Impact of 3D backbone and pre-training network alignment on performance retention across varying annotated training data sizes.

As shown in Fig. 11, there is a stark contrast in the retained representations; the view transformation method like UniPAD (Yang et al., 2024) suffers from catastrophic forgetting, producing blurry outputs that indicate a loss of fine-grained 3D information and a collapse into mean regression. In contrast, NeRP3D successfully retains structural understanding, with key elements remaining clearly distinguishable. Quantitative results in Tab. 10 further support this. While standard per-pixel metrics are sensitive to feature scale shifts induced during fine-tuning, often favoring the mean regression of UniPAD, NeRP3D significantly outperforms the view transformation method in scale-invariant (SI-RMSE) and perceptual (LPIPS, GMSD) metrics. This confirms that, unlike view transformation methods that overfit to specific tasks and collapse into mean regression, NeRP3D learns a robust and continuous representation that maintains geometric fidelity across diverse downstream objectives.

## C.2 ADAPTABILITY

We evaluate the adaptability of NeRP3D compared to the previous NeRF-based pre-training method when transferring from one detection range for pre-training to another for fine-tuning. We pre-train UniPAD (Yang et al., 2024) and our NeRP3D on the detection range optimized for 3D object detection with the full training set and subsequently fine-tune for HD map construction on a 1/2 training set. Detailed detection range for 3D object detection and HD map construction is described in Sec. 4.2.

In Fig. 5, "Map>Map" denotes that the detection range remains the same for HD map construction in both phases, while "Det>Map" indicates a change in detection range from 3D object detection during pre-training to HD map construction during fine-tuning. As a result, while view transformation-based approaches suffer substantial performance drops due to the fundamental modification of volumetric features (the size of a tensor and voxels) when changing detection range with voxel size, NeRP3D maintains consistent representation quality across different spatial configurations. This is because NeRP3D's point-based architecture only requires adjusting the coordinates of sampled points without altering the underlying representation itself. The continuous nature of our NeRF-resembled architecture highlights a key advantage of NeRP3D, namely the ability to generalize across tasks with different spatial requirements without compromising the quality of learned representations, further demonstrating the benefits of our unified point-based approach over discretized view transformation approaches.

## C.3 EFFECTIVENESS

We investigate the effectiveness of pre-training knowledge transfer in terms of the alignment of the 3D backbone and pre-training network by evaluating its performance when fine-tuned with varying amounts of annotated data. We compare the performance between UniPAD and NeRP3D when fine-tuned on 1/8, 1/4, 1/2, and the full training set.

As shown in Fig. 6, NeRP3D demonstrates robustness to reduced annotation quantities, with less performance degradation compared to UniPAD as the training set size decreases. This enhanced data efficiency can be attributed to the rich geometric and appearance information captured during pre-training, which provides a strong foundation for downstream tasks even with limited supervision. The alignment of the 3D backbone and the principle of NeRF-based pre-training enhances the

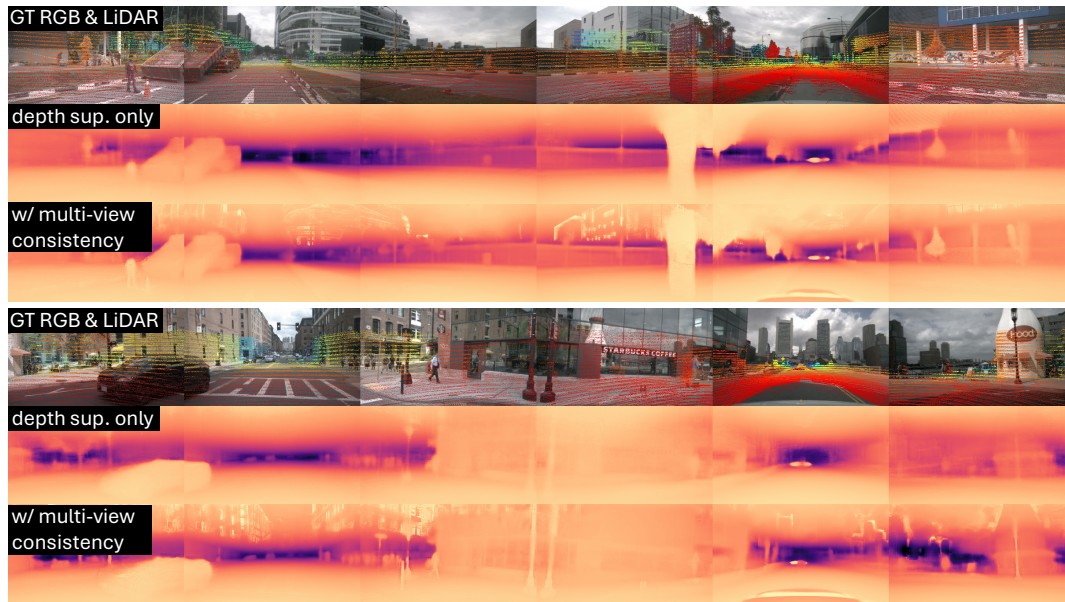

Figure 7: Qualitative comparison of depth estimation results. While LiDAR-based depth supervision alone shows limited improvement, incorporating multi-view consistency significantly enhances fine-grained and spatial accuracy, enabling plausible predictions of geometric structures even beyond the detection range.

Table 11: Ablation study on depth estimation performance with and without multi-view consistency. Sparse LiDAR scans define the *ground truth* of depth in this experiment.

| Multi-view Consistency | Abs Rel↓ | Sq Rel↓ | RMSE↓ | RMSE log ↓ | $\delta < 1.25$ ↑ | $\delta < 1.25^2$ ↑ | $\delta < 1.25^3$ ↑ |
|---|---|---|---|---|---|---|---|
| ✗ | 0.202 | 2.264 | 7.716 | 0.348 | 0.764 | 0.874 | 0.926 |
| ✓ | 0.183 | 2.274 | 7.884 | 0.353 | 0.799 | 0.883 | 0.926 |

effectiveness of knowledge transfer from pre-training to fine-tuning, allowing the model to generalize better from fewer examples in autonomous driving perception tasks.

### C.4 Depth Supervision & Multi-view Consistency

We compare two approaches for depth pre-training: relying solely on LiDAR point cloud ground truth and incorporating multi-view consistency during training. When supervision is limited to LiDAR point clouds, depth estimation is accurate within the regions covered by the sensor. However, it cannot provide meaningful predictions in areas lacking LiDAR point cloud returns. In contrast, multi-view consistency enables the model to leverage geometric cues from overlapping camera views, but it is not as accurate as LiDAR point cloud supervision.

Qualitatively, the addition of multi-view consistency provides fine-grained depth quality, allowing the model to infer plausible geometric structures in regions where LiDAR supervision is unavailable or out of range, as shown in Fig. 7. However, since depth evaluation metrics are restricted to areas with sparse LiDAR point cloud ground truth, these improvements are not fully reflected in quantitative results. In fact, as shown in Tab. 11, a model explicitly trained to optimize these evaluation metrics may achieve slightly better numerical scores on some metrics by focusing exclusively on accurate prediction at sparse LiDAR points, while potentially sacrificing overall geometric coherence and depth consistency in regions without ground truth supervision.

Furthermore, Tab. 12 demonstrates how depth supervision during pre-training impacts downstream 3D object detection. The experiment is conducted on input resolutions of $800 \times 450$ with full data. Pre-trained with only cameras using multi-view consistency, our NeRP3D model establishes a strong baseline, achieving 38.6 NDS and 34.5 mAP, which already outperforms the LiDAR-assisted UniPAD model. Moreover, incorporating LiDAR-based depth supervision during pre-training further enhances

Table 12: Ablation study on 3D object detection performance with and without depth supervision from LiDAR. "C" and "L" under Pre-train Modality denote camera for multi-view consistency and LiDAR for depth supervision, respectively.

| Method | Pre-train | Pre-train Modality | NDS↑ | mAP↑ |
|---|---|---|---|---|
| UVTR-C (Li et al., 2022) | UniPAD (Yang et al., 2024) | C & L | 37.1 | 33.7 |
| NeRP3D | Ours | C | 38.6 | 34.5 |
| NeRP3D | Ours | C & L | **39.2** | **35.8** |

this performance, boosting performance to 39.2 NDS and 35.8 mAP. This result demonstrates both the inherent effectiveness of the NeRP3D architecture and the significant, additive benefit of using explicit geometric priors from LiDAR to improve detection accuracy.

## C.5 DESIGN VALIDATION

All experiments to validate our design choice were conducted on a 1/4 subset with $704 \times 256$ images.

**Consistency of 3D Point Priors.** To verify the importance of our unified architecture rather than a point-based architecture, we applied rendering pre-training strategy to PETRv2 (Liu et al., 2023a), a point-based architecture. While PETRv2 learned 3D representations from pre-training, the performance failed to transfer to detection (approx. 0.0 mAP). This failure stems from a fundamental disruption in the consistency of 3D point priors. In pre-training, queries represent specific spatial locations to encode geometry and radiance. However, PETR's fine-tuning forces a drastic shift where queries act as object instances, invalidating the learned spatial priors. In contrast, NeRP3D maintains consistent spatial point queries across tasks, ensuring effective knowledge transfer (20.70 mAP).

**SDF Prior.** We evaluated the impact of the geometric prior by replacing SDF (NeuS (Wang et al., 2021)) with standard density (NeRF (Mildenhall et al., 2021)). The SDF-based model achieved 20.70 mAP, outperforming the density-based variant (19.35 mAP), since SDF enforces a hard surface constraint and creates sharp and unambiguous object boundaries. This structural clarity is critical for localizing and distinguishing precise objects in perception tasks, validating our design choice of using NeuS over standard NeRF.

**Deformable vs. Standard Attention.** We validated the effectiveness of deformable attention against standard global attention. Deformable attention achieved 20.70 mAP, significantly surpassing standard attention (18.38 mAP). Since each 3D point corresponds to a specific physical location, attending to the global context (standard attention) introduces irrelevant noise. Deformable attention enforces a necessary locality inductive bias by restricting the receptive field to the projected neighborhood. This proves that focusing on relevant local features is essential for accurate continuous representation.

## D COMPUTATION ANALYSIS

We evaluated the practicality and scalability by varying sampling densities. Tab. 13 demonstrates that NeRP3D operates at a computational level comparable to Uni-PAD (Yang et al., 2024) while delivering enhanced performance. Crucially, detection accuracy scales linearly with sampling density. This scalability allows the model to be tuned for performance or efficiency, depending on the resource budget.

Table 13: **Computation analysis**

| Method | GFLOPS | FPS | mAP |
|---|---|---|---|
| UniPAD | 1250.10 | 5.59 | 19.12 |
| NeRP3D | 1903.77 | 4.47 | 20.70 |
| | 1621.35 | 4.91 | 19.69 |
| | 1492.60 | 5.25 | 19.20 |
| | 1315.03 | 5.54 | 18.89 |

## E THE USE OF LARGE LANGUAGE MODELS

During the preparation of this paper, we utilized publicly available large language models (LLMs) only to aid in polishing the writing. The model's role was strictly limited to improving grammar, refining sentence structure, and enhancing the overall clarity and readability of the text. All scientific contributions, including the core ideas, experimental design, and analysis of results, are exclusively our work. We carefully reviewed and edited all model-generated suggestions and retain full responsibility for the final content of this paper.

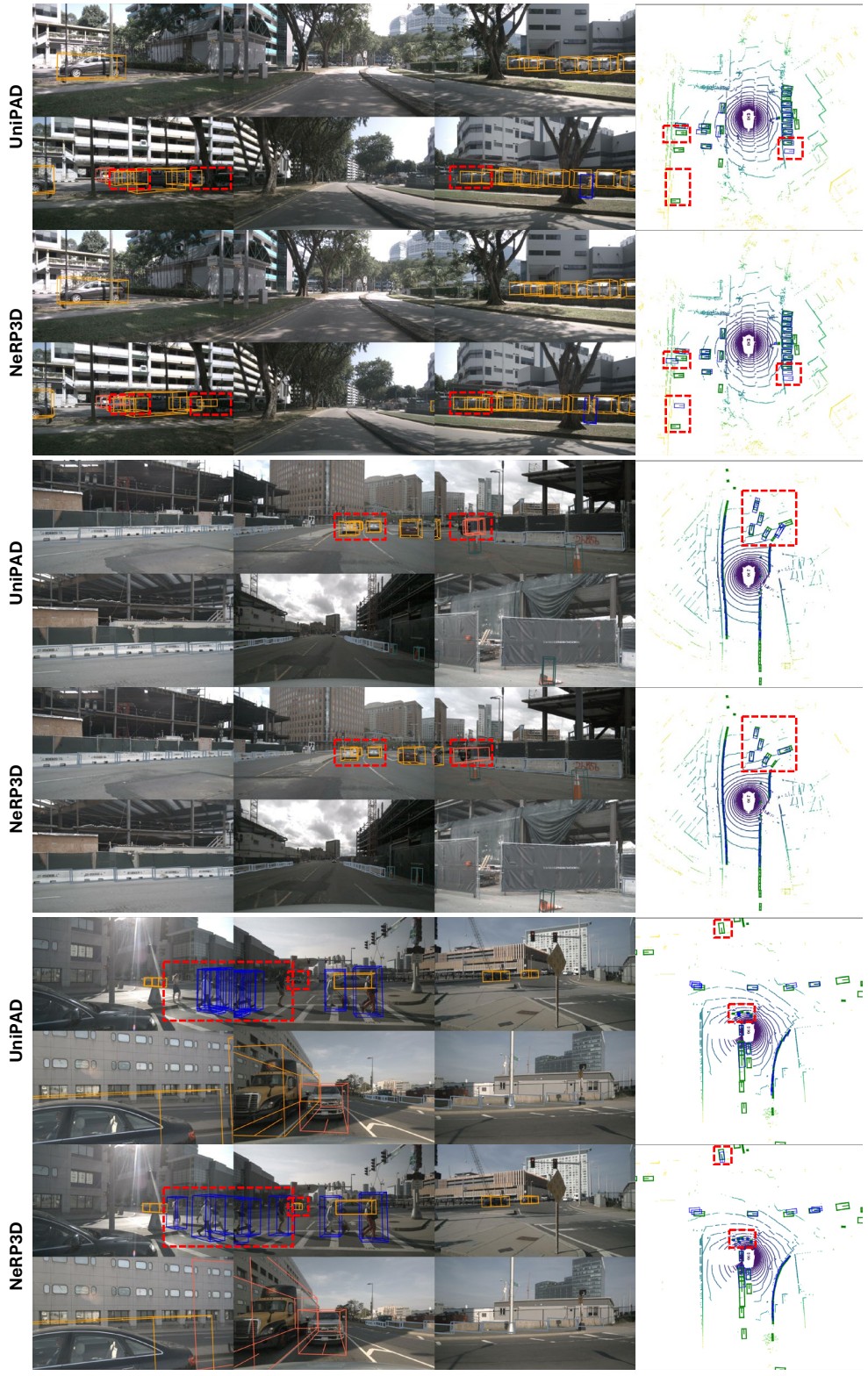

Figure 8: Qualitative comparison of 3D object detection results. NeRP3D consistently generates more accurate and reliable 3D bounding boxes. It demonstrates key advantages such as successfully detecting partially occluded objects in dense crowds (top row), reducing false positives for cleaner predictions (middle row), and more accurately localizing the position of small objects like pedestrians (bottom row).

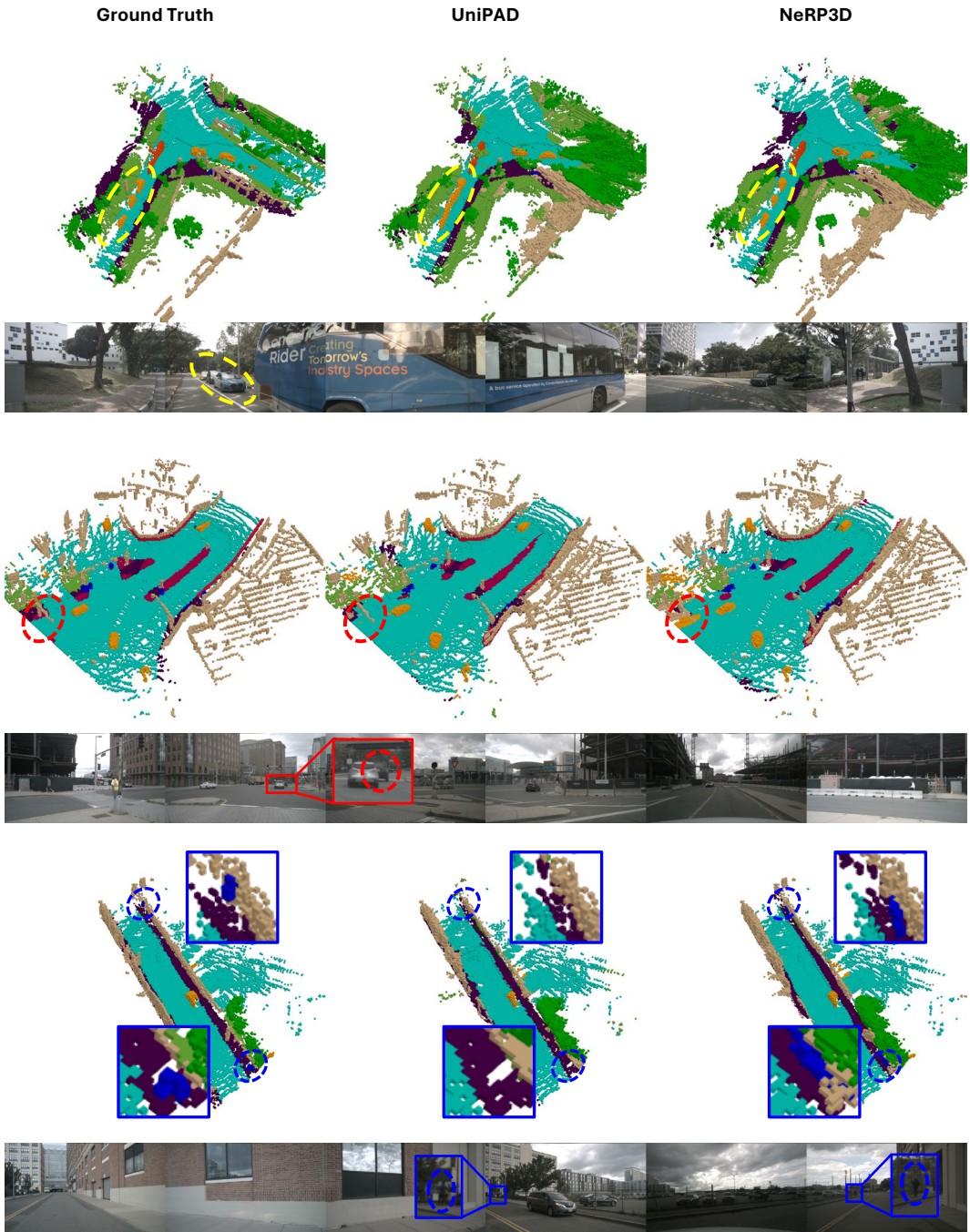

Figure 9: Qualitative comparison of occupancy prediction results. NeRP3D produces more detailed and complete occupancy predictions compared to UniPAD. NeRP3D excels at distinguishing individual objects that are close together, as shown by its clear separation of the vehicles (top row, yellow). Furthermore, it demonstrates a superior ability to detect objects that are entirely missed in the ground truth annotation, likely due to occlusion (middle row, red). The robust perception ability of NeRP3D also extends to resolving smaller, distant objects, such as pedestrians (bottom row, blue), contributing to more accurate and reliable scene understanding.

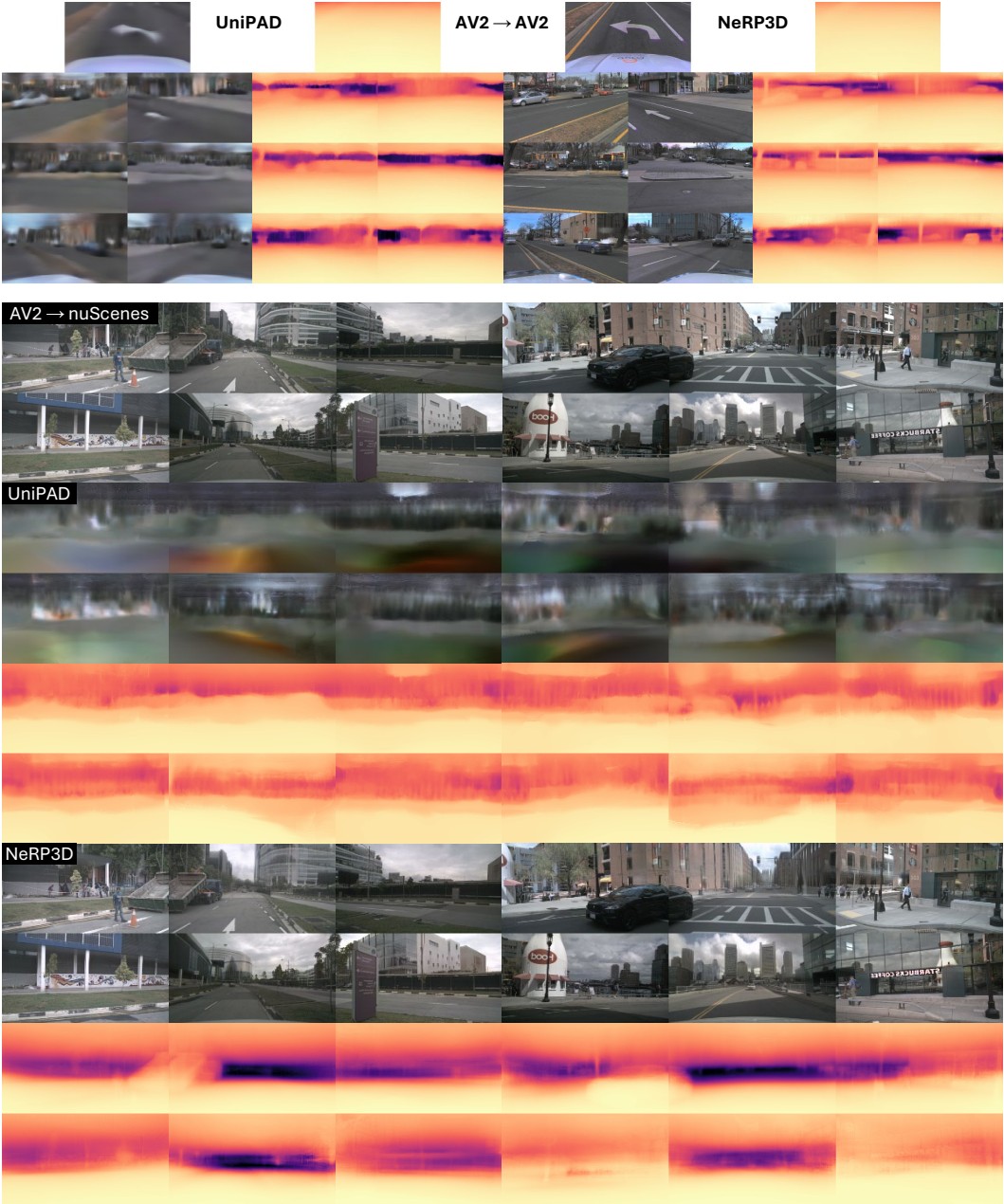

Figure 10: Qualitative comparison of zero-shot transfer for reconstruction from Argoverse 2 to nuScenes dataset (pre-training phase). Models were pre-trained on Argoverse 2 (AV2) and evaluated without any fine-tuning on the target domain (nuScenes). (Top: AV2 → AV2) In-domain reconstruction results. Both models demonstrate that pre-training on AV2 was successful, reconstructing scene details within the source domain. (Bottom: AV2 → nuScenes) Zero-shot transfer results to nuScenes. When applying AV2-trained weights directly to the distinct camera geometry of nuScenes, UniPAD fails to render meaningful structure (blurry artifacts), revealing the vulnerability of fixed voxel grids to sensor layout changes. In contrast, NeRP3D maintains high-fidelity rendering, demonstrating that its point-based architecture is sensor-agnostic and robust to extrinsic shifts.

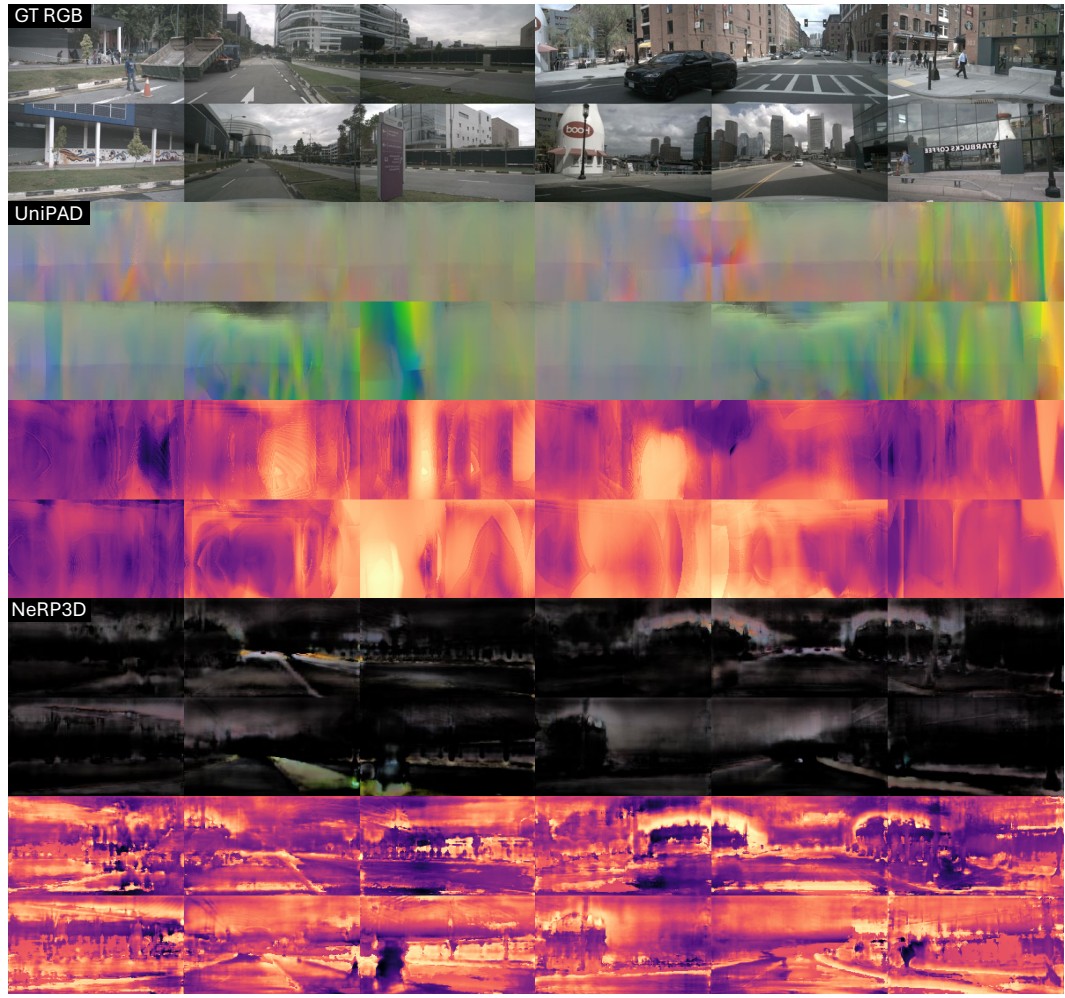

Figure 11: Qualitative evaluation of radiance modeling ability after fine-tuning for occupancy prediction. We visualize rendering results using backbones fine-tuned for the occupancy task, with pre-trained decoders frozen. (Row 2-3) UniPAD suffers from catastrophic forgetting, producing "blurry gray" outputs. The model loses 3D structural information and resorts to mean regression to minimize loss. (Row 4-5) NeRP3D successfully retains structural understanding despite the task shift. Key elements like vehicles, road boundaries, and building structures remain clearly distinguishable in both RGB and Depth renderings. This proves that NeRP3D learns a task-agnostic continuous representation that remains valid across different downstream objectives.