# OpenReview forum: "To View Transform or Not to View Transform: NeRF-based Pre-training Perspective"
_ICLR.cc/2026/Conference — ICLR 2026 Poster_

### Official Review · Reviewer_4CMH · 2025-10-28

**Soundness:** 2
**Presentation:** 2
**Contribution:** 2
**Rating:** 6
**Confidence:** 2

**Summary:**

The paper proposes NeRP3D, a new framework that rethinks how NeRF-based pre-training is used for 3D perception. It argues that existing methods with view transformations (e.g., BEV or voxel space), which introduces a mismatch between NeRF’s continuous radiance fields and discrete voxel priors. NeRP3D removes this step entirely and directly models continuous 3D points, fusing multi-view image features through deformable attention. The same network is used for both pre-training (via RGB and signed distance function) and downstream tasks like 3D detection and occupancy prediction. Experiments on nuScenes show clear improvements across all benchmarks and much sharper scene reconstructions.

**Strengths:**

1. The paper identifies a conceptual flaw in current NeRF-based pre-training methods and proposes a simple but elegant fix. Removing view transformation aligns the training process with the continuous nature of NeRFs.
2. The unified architecture is clean and consistent, avoiding the typical “throwaway pre-training” stage found in earlier works.
3. Experimental results are strong and consistent. The performance gains are not marginal, especially in image reconstruction and occupancy prediction.

**Weaknesses:**

1. The paper is conceptually motivated by the mismatch between discrete view-transformed features and continuous NeRF representations. However, it does not provide direct quantitative evidence demonstrating how this conflict affects model performance or how NeRP3D resolves it. Including controlled experiments that isolate this factor, such as measuring representational smoothness or reconstruction error across discretization levels—would strengthen the argument and make the theoretical claim more concrete.
2. The paper lacks an evaluation of runtime, computational cost, or scalability. Because NeRP3D relies on dense point sampling and deformable cross-attention, it may have higher memory and compute overhead compared to voxel-based methods. Reporting training and inference times, FLOPs, and GPU memory usage, along with a discussion of potential optimization strategies (e.g., sampling reduction or pruning), would help clarify whether the approach is practical for real-time or large-scale autonomous driving applications.
3. All experiments are conducted solely on the nuScenes dataset. While the results there are strong, it remains unclear how well NeRP3D generalizes to different environments, sensor setups, or data distributions. Evaluating the model on additional benchmarks such as Waymo Open Dataset or at least providing cross-dataset transfer results, would greatly enhance the paper’s credibility and demonstrate the robustness of the proposed representation across domains.

**Questions:**

1. How does NeRP3D compare in terms of training time and inference speed to existing voxel-based models?
2. How sensitive is the model to the number and distribution of sampled points during training?
3. Is there any evidence that the pre-trained NeRF-like encoder retains its radiance modeling ability after fine-tuning for detection or mapping?
4. Could this approach be extended to incorporate LiDAR or radar inputs directly, given that it operates in continuous 3D space?

---

> ### Author Response · Authors · 2025-11-21
> **Response to Reviewer 4CMH (1/5)**
>
> We greatly appreciate your time and effort in reviewing our work. Your insights guided us to conduct meaningful experiments that yielded new findings, significantly strengthening the quality of our paper. The answers for the weakness and your questions have been addressed below and we strongly recommend you to **check newly added supplementary PDF** which will be helpful to address your concerns.
>
> **W1. Quantitative evidence of "mismatch"**
> > W1. The paper is conceptually motivated by the mismatch between discrete view-transformed features and continuous NeRF representations. However, it does not provide direct quantitative evidence demonstrating how this conflict affects model performance or how NeRP3D resolves it. Including controlled experiments that isolate this factor, such as measuring representational smoothness or reconstruction error across discretization levels—would strengthen the argument and make the theoretical claim more concrete.
>
> We agree that measuring reconstruction error across different discretization levels is crucial, so we provide direct quantitative evidence through a "Multi-Resolution Reconstruction Analysis," which serves as the controlled experiment requested. The results isolate the discretization factor, proving that the discretization of voxel-based methods acts as a "low-pass filter" while NeRP3D preserves high-frequency details from continuous representation.
>
> **1. Quantitative Evidence: Resolution Sensitivity Analysis**
>
> We evaluated rendering performance across multiple image scales (1/16, 1/8, and 1/4).
>
> |Method|Scale|PSNR &uarr;|SSIM &uarr;|LPIPS &darr;|
> |:---------|:------:|:----------------:|:--------------:|:----------------:|
> |UniPAD|1/16 |            23.55|          0.796|            0.250|
> |UniPAD|  1/8 |            22.49|          0.664|            0.442|
> |UniPAD|  1/4 |            21.14|          0.549|            0.634|
>
> |Method|Scale|PSNR &uarr;|SSIM &uarr;|LPIPS &darr;|
> |:---------|:------:|:----------------:|:--------------:|:----------------:|
> |NeRP3D|1/16|            26.00|         0.862|             0.116|
> |NeRP3D| 1/8 |            29.40|         0.919|             0.090|
> |NeRP3D| 1/4 |            33.42|         0.969|             0.070|
>
> - Voxel-based methods (UniPAD) show inflated metrics at lower resolutions (1/16) but degrade significantly at high resolutions (1/4). This confirms that discrete voxel grids act as a **low-pass filter**. They perform well when matching blurred, low-resolution ground truth but fail to scale to high-fidelity inputs due to the rigid discretization bottleneck.
>
> - In contrast, NeRP3D maintains superior fidelity at high resolutions, with the performance gap widening as resolution increases. The slight metric drop at lower resolutions (1/16) is attributable to aliasing artifacts. Since NeRP3D predicts sharp, high-frequency details from continuous coordinates, it mismatches the smoothed/blurred ground truth at coarse scales. This pattern quantitatively proves that our architecture successfully resolves the mismatch, modeling fine-grained geometry that fixed grids cannot capture.
>
> Furthermore, we believe our existing experiments serve as the controlled tests requested, isolating the effects of discretization, smoothness, and feature resolvability.
>
> **2. Impact on Feature Resolvability (Fig. 1)**
>
> The consequence of this mismatch is visualized in Figure 1. While view transformation methods produce "blurry blobs" where distinct objects collapse, NeRP3D preserves sharp boundaries. Furthermore, NeRP3D distinguishes individual people
> in crowded areas rather than merging them into indistinct blobs as shown in Fig. 1, 4, and 8. This resolvability directly explains our superior performance in detecting crowded and small objects (e.g., pedestrians), validating that resolving the mismatch translates to tangible perception gains.
>
> **3. Impact of Discretization (Fig. 5, Adaptability)**
>
> Our "Adaptability" experiment acts as a stress test for discretization. When the voxel grid definition is altered, the VT-based method (UniPAD) suffers a significant drop (-3.25 mAP) due to its rigid priors, whereas NeRP3D remains robust (-2.32 mAP). This quantitatively proves the fragility of coupling NeRF with a discrete view transformation.

---

> ### Author Response · Authors · 2025-11-21
> **Response to Reviewer 4CMH (2/5)**
>
> **W2, Q1, Q2 Computation**
> > W2. The paper lacks an evaluation of runtime, computational cost, or scalability. Because NeRP3D relies on dense point sampling and deformable cross-attention, it may have higher memory and compute overhead compared to voxel-based methods. Reporting training and inference times, FLOPs, and GPU memory usage, along with a discussion of potential optimization strategies (e.g., sampling reduction or pruning), would help clarify whether the approach is practical for real-time or large-scale autonomous driving applications.)
>
> > Q1. How does NeRP3D compare in terms of training time and inference speed to existing voxel-based models?
>
> > Q2.How sensitive is the model to the number and distribution of sampled points during training?
>
> We provide the quantitative analysis in the table below. Contrary to the concern regarding potential high overhead, our results demonstrate that NeRP3D operates at a computational level comparable to voxel-based baselines, confirming its immediate feasibility for real-time applications. (Training time was excluded due to the difficulty of perfectly controlling data transfer latency on a shared server cluster. We prioritized metrics that could be measured with high precision in our experiments, specifically GFLOPS, FPS, and model memory.)
>
> |Method  |GFLOPS|FPS| Memory | mAP|
> |:-----------|:----------:|:-----:|:-----------:|:------:|
> |UniPAD  |1250.10 |5.59 |66.82M   |19.12|
> |NeRP3D|1903.77 |4.47 |70.93M   |20.70|
> |NeRP3D|1621.35 |4.91 |70.93M   |19.69|
> |NeRP3D|1492.60 |5.25 |70.93M   |19.20|
> |NeRP3D|1315.03 |5.54 |70.93M   |18.89|
> |NeRP3D|1156.18 |5.82 |70.93M   |18.52|
>
> First, we address the concern that the point-based architecture might incur prohibitive overhead. Our analysis reveals that the point-based architecture is not inherently prohibitive; rather, it is practical for deployment, compared to voxel-based methods, even in its high-performance state.
>
> Furthermore, regarding optimization strategies, our scalability analysis empirically validates that "sampling reduction" is an effective method for scenarios requiring stricter latency. By simply adjusting the sampling density, the model achieves higher FPS while maintaining competitive accuracy. This demonstrates that our architecture is not only practical in its default high-performance mode but also offers the flexibility to be further optimized.
>
> Finally, regarding the model's sensitivity to the number of sampled points, the above table serves as direct empirical evidence for scalability. We observe a consistent, positive correlation where increasing point density linearly translates to performance gains (18.52 $\rightarrow$ 20.70 mAP). This predictable scaling behavior indicates that the model is not unstably sensitive to sampling distributions; rather, it exhibits robust scalability, effectively utilizing more fine-grained geometric information to refine perception. This stability allows for reliable performance tuning tailored to the target application's computational budget.

---

> ### Author Response · Authors · 2025-11-21
> **Response to Reviewer 4CMH (3/5)**
>
> **W3. Generalization**
>
> > W3. All experiments are conducted solely on the nuScenes dataset. While the results there are strong, it remains unclear how well NeRP3D generalizes to different environments, sensor setups, or data distributions. Evaluating the model on additional benchmarks such as Waymo Open Dataset or at least providing cross-dataset transfer results, would greatly enhance the paper’s credibility and demonstrate the robustness of the proposed representation across domains.
>
> Following your constructive suggestion, we performed a cross-dataset evaluation to verify the robustness of our representation across different environments, sensor setups, and data distributions, demonstrating superior domain robustness and credibility.
> We conducted two experiments: (1) AV2 pre-training and nuScenes fine-tuning for 3D object detection and (2) AV2 pre-training and nuScenes zero shot rendering (**Figure 10 in newly added supplementary PDF**). Please refer to the response to Reviewer 6P5j W2 (2/4) for detailed experimental settings and the quantitative results for two experiments are reported as below:
>
> **(1) AV2 pre-training &rarr; nuScenes fine-tuning for 3D object detection**
> |             | mAP | NDS |
> |-----------|:------:|:-------:|
> |UniPAD | 26.29| 29.28|
> |NeRP3D|28.46| 31.51|
>
> **(2) AV2 pre-training &rarr; nuScenes zero shot rendering (Figure 10 in newly added supplementary PDF)**
> |               | Abs Rel &darr;| Sq Rel &darr;|RMSE &darr;|RMSE log &darr;|PSNR &uarr;|SSIM &uarr;|LPIPS &darr;|
> |:------------|:-------------------:|:-----------------:|:----------------:|:---------------------:|:---------------:|:---------------:|:----------------:|
> |UniPAD   | 0.985               | 11.767          | 14.963         | 4.390                 | 18.668         | 0.432          | 0.577          |
> |NeRP3D | 0.626               |   6.251          | 10.728         | 0.921                 | 28.238         | 0.905          | 0.111           |
>
> NeRP3D demonstrates its robustness for different environments, sensor setups, or data distributions. These results greatly enhance the credibility of our approach, proving that the continuous 3D representations learned by NeRP3D are not overfitted to a specific dataset but are robust features transferable across varying domains and distributions.
>
> Moreover, following your insightful suggestion Q3, we confirmed that NeRP3D retains valid 3D structural representations even after fine-tuning, demonstrating "Cross-Task Generalization." Detailed demonstration is in the next response.

---

> ### Author Response · Authors · 2025-11-21
> **Response to Reviewer 4CMH (4/5)**
>
> **Q3. Evidence for retainment of radiance modeling ability after finetuning**
>
> > Q3. Is there any evidence that the pre-trained NeRF-like encoder retains its radiance modeling ability after fine-tuning for detection or mapping?
>
> To validate our NeRF-resembled encoder retains its radiance modeling ability after fine-tuning, we conducted a direct rendering evaluation using the model weights after fine-tuning for the downstream task (occupancy prediction > reconstruction). Thanks to your insightful suggestions, we were able to make an interesting discovery. The setup of experiments is as follows:
>
> - Protocol: We used the fine-tuned backbone encoder but kept the pre-trained NeRF rendering heads (RGB/Depth Decoders) frozen. Since the decoders were not updated during fine-tuning, successful rendering relies entirely on whether the encoder retains the geometric and photometric features learned during pre-training.
>
> **1. Qualitative Analysis: Structure vs. Blur (Figure 12 in supplementary PDF)**
>
> As shown in **the supplementary Figure 12**, there are distinct differences in the retained representations:
> UniPAD’s results appear as "blurry gray blobs." The model has lost fine-grained 3D information and resorted to mean regression to minimize loss, failing to reconstruct any meaningful geometry.
> NeRP3D clearly preserves scene structures—road boundaries, vehicles, and buildings are distinguishable. This proves that our point-based encoder maintains "radiance modeling ability" and continuous 3D understanding even after being optimized for a different task.
>
> **2. Quantitative Analysis: Robustness over Pixel-matching**
>
> We present the standard reconstruction metrics in the main manuscript. However, note that standard per-pixel metrics (e.g., PSNR, AbsRel) are sensitive to feature scale shifts that occur during fine-tuning. UniPAD achieves higher PSNR simply by predicting the mean value (green-gray), which minimizes squared error but represents failure. We first evaluate the reconstruction on standard per-pixel metric:
>
> |               | Abs Rel &darr;| Sq Rel &darr;|RMSE &darr;|RMSE log &darr;|PSNR &uarr;|SSIM &uarr;|
> |:------------|:-------------------:|:-----------------:|:----------------:|:---------------------:|:---------------:|:---------------:|
> |UniPAD   | 0.477               | 6.914            | 15.104         | 1.056                 | 11.623        |    0.283       |
> |NeRP3D | 2.192               | 12.372          | 19.459         | 1.185                 | 9.308          |    0.135       |
>
> To properly evaluate structural preservation, we employ perceptual, structural and scale-invariant metrics [1], [2]:
>
> |               | SI RMSE &darr;| Grad Loss &darr;|GMSD &darr;|LPIPS &darr;|PSNR-HM &uarr;|SSIM-HM &uarr;|
> |:------------|:--------------------:|:----------------------:|:-----------------:|:----------------:|:---------------------:|:---------------:|
> |UniPAD   | 0.859                | 90.164                | 0.306            | 0.863            | 12.319               | 0.300          |
> |NeRP3D | 0.643                 | 83.739                | 0.289            | 0.671            | 12.839              | 0.285          |
>
> Although NeRP3D performs worse in the per-pixel metric than expected based on qualitative results, it performs well overall in structural and perceptual metrics.
> NeRP3D achieves a significantly lower score for perceptual loss, LPIPS (0.67 vs. 0.86), indicating superior perceptual quality and less blur. Furthermore, regarding GMSD (Gradient Magnitude Similarity Deviation), which evaluates the structural integrity of the image based on local gradient changes, NeRP3D achieves a superior score (0.289 vs. 0.306). For SI-RMSE (Scale-Invariant), NeRP3D outperforms UniPAD (0.64 vs. 0.86), proving that the geometric depth structure is preserved despite the absolute scale shift. Moreover, NeRP3D shows lower gradient loss, indicating better preservation of edges and shapes.
>
> In conclusion, while fine-tuning induces a feature scale shift (affecting PSNR/AbsRel), NeRP3D successfully retains the structural and semantic radiance modeling ability, unlike VT-based methods that suffer from catastrophic forgetting of the 3D scene structure.
>
> [1] Eigen et al. (2014), Depth Map Prediction from a Single Image using a Multi-Scale Deep Network
>
> [2] Xue et al. (2014), Gradient Magnitude Similarity Deviation: A Highly Efficient Perceptual Image Quality Index.

---

> ### Author Response · Authors · 2025-11-21
> **Response to Reviewer 4CMH (5/5)**
>
> **Q4. Incorporation with LiDAR or radar**
> > Q4. Could this approach be extended to incorporate LiDAR or radar inputs directly, given that it operates in continuous 3D space?
>
> The continuous point-based nature of NeRP3D makes it exceptionally well-suited for integrating sparse geometric inputs like LiDAR or radar. We expect that this extension brings significant benefits in efficiency and performance due to the three expectations below:
>
> **1. Efficiency via Guided Sampling**
>
> - Current constraint: NeRP3D relies on volumetric ray marching to discover geometry from scratch, which requires sampling a large number of points across the 3D space.
> - Expected Extension: Incorporating LiDAR or radar would provide strong geometric priors. Instead of blind volumetric sampling, we could constrain the query points to the vicinity of the sensor measurements. As demonstrated by methods like GaussianPretrain [3], using such geometric initialization drastically reduces the number of required sampling points, significantly boosting computational efficiency.
>
> **2. Enabling Richer Representations**
>
> The computational budget saved by guided sampling could be reallocated to model richer point attributes. For instance, instead of simple RGB/density, the model could learn more complex primitives (e.g., 3D Gaussians or higher-order harmonics) at the localized points, enabling higher-fidelity scene representation without increasing total latency.
>
> **3. Performance Enhancement**
>
> Directly incorporating explicit depth measurements would help resolve the scale ambiguity inherent in vision-only methods, leading to more deterministic object localization and superior detection performance.
>
> [3] Xu et al. (2024), GaussianPretrain: A Simple Unified 3d Gaussian Representation for Visual Pre-training in Autonomous Driving

---

### Official Review · Reviewer_wj2C · 2025-10-29

**Soundness:** 3
**Presentation:** 3
**Contribution:** 2
**Rating:** 4
**Confidence:** 3

**Summary:**

This paper proposes NeRP3D, a NeRF-Resembled Point-based 3D detector for autonomous driving that avoids view transformation by directly modeling continuous 3D representations. The authors argue that existing NeRF-based pre-training methods (UniPAD, SelfOcc) suffer from conflicting priors: view transformation imposes discrete/rigid representations while NeRFs assume continuous functions. NeRP3D preserves the pre-trained NeRF network during downstream tasks and supports adaptive sampling strategies (ray-wise for rendering, spatial for detection). Experiments on nuScenes show modest gains ($1-2\%$) over UniPAD in 3D detection, occupancy, and HD map tasks.

**Strengths:**

1. Strong Problem Diagnosis. The paper clearly identifies the fundamental tension between discrete View Transformation and continuous NeRF priors. The critique of this "conflicting prior" (Fig 1) is insightful and provides a solid motivation.

2. Architectural Coherence. The unified framework for pre-training (ray-wise sampling) and downstream tasks (spatial sampling) is elegant. It effectively preserves the pre-trained NeRF representation without an intermediate discrete stage.

3. Comprehensive experimental validation. The paper evaluates across multiple pretext tasks (RGB/depth reconstruction) and three downstream tasks

**Weaknesses:**

1. Limited Methodological Novelty. The core idea of querying 2D features at 3D points to maintain a continuous representation is not new (e.g., pixelNeRF [1], MVSNeRF [2]). The contribution is an architectural choice (avoiding an intermediate voxel grid) rather than a fundamental new modeling approach.

2. Insufficient Ablations. Key design choices are unjustified. (a) Why deformable cross-attention (Eq 2) over standard attention? (b) The point sampling density trade-off (accuracy vs. cost) is not analyzed.

3. Incomplete Cost Analysis. The paper acknowledges high costs (line 484) but fails to quantify them. Key metrics are missing: (a) Inference latency (ms/frame) and (b) Scalability analysis (i.e., the accuracy/cost trade-off vs. the number of sampled points).

4. Missing Comparisons to Key Baselines. The paper only compares against NeRF-pretrain (UniPAD) and VT-based (BEVFormer) methods. It critically omits comparisons to point-based detectors (e.g., DETR3D[3], PETR[4]). These methods share NeRP3D's core philosophy of querying continuous 3D locations. Without this comparison, it is impossible to know if the gains come from NeRF pre-training or simply from the point-based representation itself.

5. Single-Dataset Evaluation. All experiments are confined to nuScenes. Without cross-dataset validation (e.g., Waymo, KITTI, Argoverse), the method's robustness to different sensors, environments, and camera layouts is entirely unverified.

[1] Yu, Alex, et al. "pixelnerf: Neural radiance fields from one or few images." Proceedings of the IEEE/CVF conference on computer vision and pattern recognition. 2021.

[2] Chen, Anpei, et al. "Mvsnerf: Fast generalizable radiance field reconstruction from multi-view stereo." Proceedings of the IEEE/CVF international conference on computer vision. 2021.

[3] Wang, Yue, et al. "Detr3d: 3d object detection from multi-view images via 3d-to-2d queries." Conference on robot learning. PMLR, 2022.

[4] Liu, Yingfei, et al. "Petr: Position embedding transformation for multi-view 3d object detection." European conference on computer vision. Cham: Springer Nature Switzerland, 2022.

**Questions:**

Please see the critical issues and required clarifications detailed in the Weaknesses section.

---

> ### Author Response · Authors · 2025-11-21
> **Response to Reviewer wj2C (1/5)**
>
> Thank you for your insightful and constructive reviews. We deeply share the reviewer's feedbacks which are the aspects we debated during our research. We hope that our response have successfully addressed the reviewer's concerns.
>
> **W1. Limited Methodological Novelty.**
> >  W1.The core idea of querying 2D features at 3D points to maintain a continuous representation is not new (e.g., pixelNeRF [1], MVSNeRF [2]). The contribution is an architectural choice (avoiding an intermediate voxel grid) rather than a fundamental new modeling approach.
>
> Our core contribution, as implied by our title, is challenging the necessity of View Transformation and establishing the first "Unified Perception Backbone" based on continuous representation. We respectfully argue that our novelty lies not in the querying operation itself, but in the paradigm shift and task generalization tailored for large-scale autonomous driving.
>
> While we acknowledge that the point-query mechanism in NeRP3D is similar to that of PixelNeRF[1] and MVSNeRF[2], we view this operation not as a specialized methodology unique to specific works, but as a universal attention mechanism for lifting 2D features to 3D space. Our novelty lies not in inventing the querying operation itself, but in the paradigm shift of applying this mechanism to establish a grid-free perception framework. Our contributions are specifically summarized as follows:
>
> **1. Architectural Shift: "Not to View Transform" (Paradigm Shift)**
>
> As suggested by our title ("To View Transform or Not to View Transform"), our primary goal is to question and challenge the dominant View Transformation (VT) paradigm. Existing methods force continuous NeRF priors into discrete voxel grids, leading to the "conflicting priors" described in the paper. NeRP3D demonstrates that a grid-free, continuous architecture (i.e., choosing not to view transform) is essential for maximizing the quality of pre-trained representations. This architectural choice is not merely an alternative but a necessary evolution to prevent information loss during pre-training.
>
> **2. Methodological Extension: From Visual Reconstruction to Semantic Perception**
>
> While prior works like PixelNeRF focus on reconstructing RGB and density for novel view synthesis, we extend this methodology to discriminative perception tasks. We demonstrate for the first time that individual 3D point queries can serve as carriers of rich semantic features (not just color) capable of supporting complex downstream tasks like 3D detection and occupancy prediction. This generalizes the utility of "point-based querying" beyond computer graphics to computer vision.
>
> **3. Foundational Framework for Future Perception Backbone, Neural Fields**
>
> NeRP3D establishes a minimal and effective baseline for applying continuous 3D representation learning to large-scale autonomous driving. By proposing a streamlined, NeRF-resembled architecture that removes the dependency on complex VT modules, we provide a flexible **Plug-and-Play** framework where future advancements in Neural Fields (e.g., 3D Gaussian Splatting) can be seamlessly integrated, paving the way for the next generation of perception models.

---

> ### Author Response · Authors · 2025-11-21
> **Response to Reviewer wj2C (2/5)**
>
> **Q2-A. Design choice for deformable attention**
> > Q2. Insufficient Ablations: Key design choices are unjustified. (a) Why deformable cross-attention (Eq 2) over standard attention?
>
> We chose deformable attention because its **local inductive bias** aligns perfectly with our core philosophy of "continuous and fine-grained 3D representation." While efficiency is a benefit, the primary motivation conceptual aligns with our point-based rendering framework. We justified in terms of distinct feature between deformable and standard attention and demonstrate our philosophy with quantitative result as below.
>
> Similar to NeRF, our method represents the scene through a set of 3D points, where **each point is responsible for encoding the detailed geometry and radiance of its specific physical location** rather than entire space.
>
> However, standard attention forces each 3D point to aggregate information from the entire image context. Since a specific 3D point needs to correspond to a precise local region in the 2D view, attending to the **global context** introduces significant noise from irrelevant regions, diluting the fine-grained local details necessary for continuous representation.
>
> On the other hand, **deformable attention enforces a locality inductive bias by restricting the receptive field to the neighborhood of the point's projection.** This allows each point to focus intensively on its spatially corresponding features, enabling the fine-grained exploration for continuous space.
>
> This conceptual alignment translates directly into performance. As shown in the ablation study below. Deformable attention outperforms standard attention by a significant margin of +2.32 mAP (20.70 vs. 18.38).
>
> |               |Complexity |mAP |
> |:------------|:--------------:|:-----:|
> |Standard|O(NqHWC)|18.38|
> |Deformable|O(NqC)|20.70|

---

> ### Author Response · Authors · 2025-11-21
> **Response to Reviewer wj2C (3/5)**
>
> **Q2-B, Q3. Computation**
> > Q2. Insufficient Ablations: (b) The point sampling density trade-off (accuracy vs. cost) is not analyzed.
>
> > Q3. Incomplete Cost Analysis. The paper acknowledges high costs (line 484) but fails to quantify them. Key metrics are missing: (a) Inference latency (ms/frame) and (b) Scalability analysis (i.e., the accuracy/cost trade-off vs. the number of sampled points).
>
> We conducted a detailed scalability analysis (Table R4), confirming that NeRP3D exhibits excellent scalability and achieves superior performance with comparable inference latency.
> We provide the requested metrics—inference latency (ms/frame) and the accuracy/cost trade-off analysis—in the table below.
>
> |Method  |GFLOPS|FPS|Model Size | mAP|
> |:-----------|:----------:|:-----:|:-----------:|:------:|
> |UniPAD  |1250.10 |5.59 |66.82M   |19.12|
> |NeRP3D|1903.77 |4.47 |70.93M   |20.70|
> |NeRP3D|1621.35 |4.91 |70.93M   |19.69|
> |NeRP3D|1492.60 |5.25 |70.93M   |19.20|
> |NeRP3D|1315.03 |5.54 |70.93M   |18.89|
> |NeRP3D|1156.18 |5.82 |70.93M   |18.52|
>
> The results illustrate a clear positive correlation between computational investment and detection accuracy, verifying the scalability of our architecture. As the number of sampled points increases, the model consistently yields higher mAP (18.52 $\rightarrow$ 20.70). This linear scaling behavior demonstrates that NeRP3D effectively utilizes additional computational resources to refine geometric representation and enhance perception performance.
>
> Furthermore, even under limited computational budgets, NeRP3D maintains performance comparable to voxel baselines. Crucially, unlike fixed grids, its continuous nature unlocks superior scalability. Therefore, NeRP3D provides a robust foundation that performs reliably under constrained resources while offering the unique capability to leverage additional compute for fine-grained spatial understanding and higher accuracy when available.

---

> ### Author Response · Authors · 2025-11-21
> **Response to Reviewer wj2C (4/5)**
>
> **W4. Missing Comparisons to Key Baselines.**
> > W4. The paper only compares against NeRF-pretrain (UniPAD) and VT-based (BEVFormer) methods. It critically omits comparisons to point-based detectors (e.g., DETR3D[3], PETR[4]). These methods share NeRP3D's core philosophy of querying continuous 3D locations. Without this comparison, it is impossible to know if the gains come from NeRF pre-training or simply from the point-based representation itself.
>
> We agree with the concern about whether the benefits come from NeRF-resembled or point-based architecture. To verify if our gains stem merely from the point-based architecture or our specific design, we applied NeRF pre-training strategy to PETRv2.
>
> **1. Experimental results: Successful pre-training, failed transfer**
>
> As shown **Figure 11 (Attached supplementary PDF)**, the pre-trained PETR backbone learned 3D representations (comparable to UniPAD). However, this knowledge failed to transfer to the detection task, resulting in near-zero performance (~0.0 mAP).
>
> |Method |Pre-training Query|Fine-tuning Query|mAP |
> |:----------|:-----------------------:|:----------------------:|:------:|
> |UniPAD|Voxel Grid             | Voxel Grid           | 19.12|
> |PETRv2|-                            | Latent Instance   |17.81|
> |PETRv2+NeRF|Explicit 3D Point|Latent Instance|~0.0|
> |NeRP3D(Ours)|Explicit 3D Point|Explicit 3D Point|20.70|
>
> **2.Conclusion: Consistency of 3D Point Priors**
>
> This comparison demonstrates the root cause of the performance gap, architectural mismatch. While PETR switches from explicit 3D point queries to latent instance queries, which also stems for candidates of physical objects, during fine-tuning, this drastic shift invalidates the learned geometric priors, leading to optimization failure. In contrast, NeRP3D maintains consistent explicit point queries across both stages.
>
> Therefore, the superior performance of NeRP3D is not a byproduct of simply using a point-based architecture (as PETR failed), but a direct result of our unified design that seamlessly aligns the NeRF mechanism, enabling effective transfer of radiance modeling abilities to downstream perception.

---

> ### Author Response · Authors · 2025-11-21
> **Response to Reviewer wj2C (5/5)**
>
> **W5. Single-Dataset Evaluation**
> > W5. All experiments are confined to nuScenes. Without cross-dataset validation (e.g., Waymo, KITTI, Argoverse), the method's robustness to different sensors, environments, and camera layouts is entirely unverified.
>
> We agree with the concern about single-dataset overfitting problem. To address the concern regarding robustness to different sensor layouts and environments, we utilized Argoverse 2, which possesses distinct camera geometries and environmental statistics compared to nuScenes but also provides 360 deg camera views. Two experiments, (1) AV2 pre-training and nuScenes fine-tuning for 3D object detection and (2) AV2 pre-training and nuScenes zero shot rendering (**Figure 10 in newly added supplementary PDF**), conducted on Argoverse 2 to nuScenes confirm that NeRP3D is robust to sensor layout changes, significantly outperforming voxel-based methods in domain transfer. Please refer to the response to **Reviewer 6P5j W2 (2/4)** for detailed experimental settings and the quantitative results for two experiments are reported as below:
>
> **(1) AV2 pre-training &rarr; nuScenes fine-tuning for 3D object detection**
> |             | mAP | NDS |
> |-----------|:------:|:-------:|
> |UniPAD | 26.29| 29.28|
> |NeRP3D|28.46| 31.51|
>
> **(2) AV2 pre-training &rarr; nuScenes zero shot rendering (Figure 10 in newly added supplementary PDF)**
> |               | Abs Rel &darr;| Sq Rel &darr;|RMSE &darr;|RMSE log &darr;|PSNR &uarr;|SSIM &uarr;|LPIPS &darr;|
> |:------------|:-------------------:|:-----------------:|:----------------:|:---------------------:|:---------------:|:---------------:|:----------------:|
> |UniPAD   | 0.985               | 11.767          | 14.963         | 4.390                 | 18.668         | 0.432          | 0.577          |
> |NeRP3D | 0.626               |   6.251          | 10.728         | 0.921                 | 28.238         | 0.905          | 0.111           |
>
> This result directly validates our method’s robustness. Unlike voxel-based view transformations (e.g., UniPAD) that rely on fixed ego-centric grids sensitive to sensor extrinsic changes, NeRP3D's continuous point-based query mechanism effectively handles diverse camera layouts, enabling seamless adaptation to new sensor configurations.
>
> Additionally, we provide evidence of "Internal Robustness" via Cross-Task Generalization. To further verify the robustness of our architecture, we analyzed whether the learned representations survive the optimization pressure of a different downstream task, as detailed in **our response to Reviewer 4CMH Q3 (4/5)** (**Figure 12 in supplementary PDF**).
>
> Our experiments reveal that while voxel-based methods suffer from catastrophic forgetting (collapsing to mean values) during fine-tuning, NeRP3D preserves clear 3D geometric structures (as evidenced by successful rendering with fine-tuned weights).
> This "Cross-Task" consistency confirms that our continuous point-based architecture is inherently robust. It maintains stable and valid 3D representations not only against sensor shifts (as shown in the Argoverse experiment) but also against drastic changes in task objectives.

---

### Official Review · Reviewer_6P5j · 2025-11-01

**Soundness:** 3
**Presentation:** 3
**Contribution:** 3
**Rating:** 6
**Confidence:** 4

**Summary:**

This paper critiques existing NeRF-based pre-training methods in autonomous driving due to their reliance on rigid view transformations, creating conflicts with NeRF's continuous representation capabilities. The authors then introduce NeRP3D, a point-based 3D detector that retains continuous NeRF representation from pre-training through downstream tasks. Experimental results on the nuScenes dataset demonstrate considerable improvements in both reconstruction quality and detection performance compared to prior methods.

**Strengths:**

1. The paper's primary strength is its core insight. Identifying the "misaligned prior" between discrete view-transformation and continuous radiance fields is a valuable critique of existing methods. This moves beyond simply "adding NeRF" and asks a more fundamental question about how it should be integrated.

2. The proposed NeRP3D architecture is an elegant solution to the problem it identifies. The unification of the pre-training and fine-tuning architectures into a single, continuous, point-queryable function is conceptually clean. This design, which preserves the pre-trained network rather than discarding it, is a good contribution to the field of 3D pre-training.

3. The experiments are solid and provide comprehensive evidence for the paper's claims.

**Weaknesses:**

Overall I think this is a good work. Only some minor weaknesses remain.

1. Computational Cost: The paper admits in the conclusion that the point-based architecture "incurs high computational costs". However, the paper provides no quantitative analysis (e.g., FLOPS, latency, memory) to compare NeRP3D against the view-transform-based baselines. This is a critical omission for a paper aimed at practical autonomous driving applications.

2. Generalization: While results on the nuScenes dataset are promising, evaluation on additional datasets could help establish broader applicability and robustness of the method.

3. Gains vs. the closest continuous baseline are small. Against GaussianPretrain, the margin on detection is +0.1 NDS / +1.1 mAP, which is within typical variance unless confidence intervals are shown.

**Questions:**

1. Compute trade-offs. Please provide FPS, peak memory, and point count vs. accuracy curves for NeRP3D and UniPAD/TPV across identical image resolutions and heads. This will show whether the continuous design is cost-effective at scale.

2. Influence of SDF Prior: The pre-training uses NeuS (SDF), while competitors use standard NeRF (density). How much of the performance gain, especially in geometric fidelity, is attributable to the SDF prior versus the continuous architecture? What happens if NeRP3D is pre-trained with a standard density-based NeRF loss?

---

> ### Author Response · Authors · 2025-11-21
> **Response to Reviewer 6P5j (1/4)**
>
> We thank the reviewer for the detailed and constructive feedback. We appreciate the positive remarks regarding insight for "misalignment prior" and the integration of NeRF. We also appreciate the insightful questions and concerns that can strengthen our research. We have carefully addressed all concerns below. And we prepared a supplementary PDF to help you with your review.
>
> **W1, Q1. Computation**
> > W1. Computational Cost: The paper admits in the conclusion that the point-based architecture "incurs high computational costs". However, the paper provides no quantitative analysis (e.g., FLOPS, latency, memory), demonstrating that NeRP3D offers comparable cost-effectiveness to voxel-based baselines while unlocking superior performance scalability and the benefits of continuous representation.
>
> > Q1. Compute trade-offs. Please provide FPS, peak memory, and point count vs. accuracy curves for NeRP3D and UniPAD/TPV across identical image resolutions and heads. This will show whether the continuous design is cost-effective at scale.
>
> We provide the requested quantitative analysis for computational cost as below, demonstrating that NeRP3D offers comparable cost-effectiveness to voxel-based baselines while unlocking superior performance scalability and the benefits of continuous representation. To address the concern regarding practicality, we evaluated key metrics (GFLOPS, FPS, Memory) against UniPAD under identical resolutions.
>
> |Method  |GFLOPS|FPS| Model |GPU(MiB)| mAP|
> |:-----------|:----------:|:-----:|:-----------:|:-----------:|:------:|
> |UniPAD  |1250.10 |5.59 |66.82M   |3168|19.12|
> |NeRP3D|1903.77 |4.47 |70.93M   |4158|20.70|
> |NeRP3D|1621.35 |4.91 |70.93M   |3742|19.69|
> |NeRP3D|1492.60 |5.25 |70.93M   |3602|19.20|
> |NeRP3D|1315.03 |5.54 |70.93M   |3450|18.89|
> |NeRP3D|1156.18 |5.82 |70.93M   |3198|18.52|
>
> Our analysis confirms that the continuous design is cost-effective at scale. As shown in Table R4, NeRP3D achieves 19.20 mAP (surpassing UniPAD's 19.12 mAP) with a latency of 5.25 FPS, which is highly comparable to UniPAD's 5.59 FPS. This demonstrates that NeRP3D matches the efficiency of view-transformation methods while avoiding their discretization artifacts.
>
> Beyond this parity, NeRP3D offers a significantly higher performance ceiling. By increasing the point density, we achieve a substantial gain (20.70 mAP). This demonstrates that our architecture provides a scalable trade-off, allowing users to maximize accuracy when computational resources permit.
>
> In summary, this quantitative analysis demonstrates that NeRP3D is a practical solution for autonomous driving. The results prove that our point-based architecture maintains computational efficiency comparable to voxel-based baselines while remaining cost-effective at scale due to its structural flexibility.

---

> ### Author Response · Authors · 2025-11-21
> **Response to Reviewer 6P5j (2/4)**
>
> **W2. Generalization**
> > W2. Generalization: While results on the nuScenes dataset are promising, evaluation on additional datasets could help establish broader applicability and robustness of the method.
>
> We agree that establishing broader applicability is essential. To demonstrate this, we conducted a cross-dataset transfer experiment using Argoverse 2 (AV2), which provides different environments and sensor configurations compared to nuScenes.
>
> **(1) AV2 pre-training &rarr; nuScenes fine-tuning for 3D object detection**
>
> - Protocol: We pre-trained the model on the AV2 dataset (1/4 split) and fine-tuned it for on the nuScenes’s 3D object detection dataset (1/2 split).
> - Results: NeRP3D achieved 27.46 mAP and 30.51 NDS, significantly outperforming UniPAD (26.29 mAP, 29.28 NDS).
>
> |             | mAP | NDS |
> |-----------|:------:|:-------:|
> |UniPAD |26.29| 29.28|
> |NeRP3D|27.46| 30.51|
>
> The substantial margin (+1.17 mAP) confirms that NeRP3D learns more universal geometric representations that generalize effectively across different domains, whereas voxel-based methods struggle to transfer learned priors to new environments.
>
> **(2) AV2 pre-training &rarr; nuScenes zero shot rendering**
>
> We also conducted experiments for reconstruction, pre-training on AV2 and evaluating without fine-tuning on the target domain (nuScenes). The qualitative results for this experiment are in **Figure 10 of attached supplementary PDF**. The quantitative results for renderin 1/8 images are in the table below:
>
> |               | Abs Rel &darr;| Sq Rel &darr;|RMSE &darr;|RMSE log &darr;|PSNR &uarr;|SSIM &uarr;|LPIPS &darr;|
> |:------------|:-------------------:|:-----------------:|:----------------:|:---------------------:|:---------------:|:---------------:|:----------------:|
> |UniPAD   | 0.985               | 11.767          | 14.963         | 4.390                 | 18.668         | 0.432          | 0.577          |
> |NeRP3D | 0.626               |   6.251          | 10.728         | 0.921                 | 28.238         | 0.905          | 0.111           |
>
> Furthermore, we demonstrate "Cross-Task Generalization" to validate broad applicability. Beyond generalizing across data domains, a truly robust representation should be resilient to shifts in task objectives.
> As detailed in our response to **Reviewer 4CMH Q3 (4/5)**, we found that the NeRP3D backbone retains its ability to render structural scene geometry even after being fine-tuned for a completely different task (**as shown in Figure 12 of newly added supplementary PDF**). This contrasts sharply with voxel-based baselines, which lose 3D structural information during fine-tuning.
>
> These capabilities imply that NeRP3D learns universal 3D features that are applicable across both datasets and tasks without losing their underlying geometric fidelity, further establishing the method's broad applicability.

---

> ### Author Response · Authors · 2025-11-21
> **Response to Reviewer 6P5j (3/4)**
>
> **W3. Gains vs. the closest continuous baseline are small. (GaussianPretrain)**
> > W3. Gains vs. the closest continuous baseline are small. Against GaussianPretrain, the margin on detection is +0.1 NDS / +1.1 mAP, which is within typical variance unless confidence intervals are shown.
>
> Comparing NeRP3D against GaussianPretrain is methodologically unequal. Surpassing GaussianPretrain that relies on explicit LiDAR initialization for points (Gaussian) location serves as a strong validation of our "geometry learning" capability.
>
> **1. GaussianPretrain: Explicit Initialization & Voxel Dependency (Strong Prior)**
>
> - LiDAR Initialization: LiDAR Initialization: GaussianPretrain explicitly anchors 3D Gaussians to LiDAR point clouds collected during pre-training. It bypasses the challenge of geometry discovery by assuming LiDAR points as the ground truth, optimizing only attributes like color. This is essentially "memorizing" sensor data rather than learning a representation.
>
> - Voxel Limitation: Furthermore, after fixing locations via LiDAR, it extracts features by interpolating from voxel grids (View Transformation), inheriting the discretization limitations of methods like UniPAD.
>
> **2. NeRP3D: Volumetric Learning (Zero Prior)**
>
> In contrast, NeRP3D starts with no geometric initialization. It utilizes volumetric ray marching to discover surface locations and learn density/SDF fields through self-supervision. Moreover, the integration of NeRF-resembled encoder enables the model to represent continuous field.
>
> The fact that NeRP3D outperforms GaussianPretrain—despite the competitor having the "unfair advantage" of explicit LiDAR initialization—proves that our learned continuous representation is still superior to representations that heavily rely on sensor priors. This confirms that NeRP3D is not just "fitting" to points but effectively "understanding" the 3D scene structure.

---

> ### Author Response · Authors · 2025-11-21
> **Response to Reviewer 6P5j (4/4)**
>
> **Q2. SDF Prior**
> > Q2. Influence of SDF Prior: The pre-training uses NeuS (SDF), while competitors use standard NeRF (density). How much of the performance gain, especially in geometric fidelity, is attributable to the SDF prior versus the continuous architecture? What happens if NeRP3D is pre-trained with a standard density-based NeRF loss?
>
> There is a slight misunderstanding, competitors like UniPAD and SelfOcc also utilize NeuS-based rendering as mentioned in Sec 4.3 Implementation Details, but we still agree that verifying the impact of SDF prior. To address this, we conducted an ablation study replacing NeuS with standard NeRF (density-based) in our framework.
>
> We evaluated the experiments on 1/4 dataset. The results below show the impact of the geometric prior on detection performance.
>
> |Pre-training Method|Geometric Prior|mAP|
> |:-------------------------|:--------------------:|:-----:|
> |NeRF                      | Density            |19.35|
> |NeuS                      | SDF                 | 20.70|
>
> Standard NeRF models geometry via volume density, often resulting in "foggy" or "thick" boundaries. This volumetric ambiguity makes it difficult to precisely separate objects from the background or objects from objects, which is critical to detection.
>
> In contrast, SDF models geometry as a hard surface constraint (strictly distinguishing inside from outside). This forces the model to learn sharp, unambiguous boundaries instead of density clouds. Our results indicate that this structural clarity, knowing exactly where an object ends, is more critical for object localization, explaining the superiority of the SDF prior.

---

### Author Response · Authors · 2025-11-30
**General Response: Summary of Rebuttals and Additional Experiments**

We sincerely thank the Reviewers (6P5j, wj2C, 4CMH) and the Program and Area Chairs for their time and constructive feedback. We are encouraged that the reviewers unanimously recognized the value of our work, particularly the insight regarding the **misaligned prior** between discrete view transformation and NeRF, the **unified architecture** that preserves continuous representations, and the **solid experimental evidence** supporting our claims.

Furthermore, your insightful concerns have been instrumental in identifying areas for improvement and significantly strengthening the robustness of our work. To address the raised concerns, we have conducted extensive additional experiments. We summarize the key updates and findings below. Detailed quantitative and qualitative results are provided in individual responses and the **newly attached Supplementary PDF**.

**1. Common Concern: Generalization & Robustness** (Addressed in Response to Reviewer 6P5j W2; wj2C W5; 4CMH W3)

To validate robustness beyond a single dataset, we conducted cross-dataset transfer experiments using **Argoverse 2 (AV2)**, which features distinct sensor configurations compared to nuScenes.

- **Domain Robustness**: When transferring from AV2 to nuScenes, NeRP3D significantly outperformed the voxel-based method for both downstream detection and pretext reconstruction (**Fig 10 in supplementary PDF**), demonstrating superior robustness to sensor layout shifts and environmental changes.

- **Cross-Task Generalization**: We further demonstrated with qualitative results (**Fig 12 in supplementary PDF**) and structural scores that NeRP3D retains valid 3D structural representations even after fine-tuning for downstream tasks, validating its task-agnostic representation capabilities.

**2. Common Concern: Computational Cost & Scalability** (Addressed in Response to Reviewer 6P5j W1, Q1; wj2C Q2, Q3; 4CMH W2, Q1, Q2)

Addressing the concerns regarding the scalability and computational practicality of our point-based architecture, we performed a quantitative scalability analysis (GFLOPS, FPS, Memory) by varying sampling densities.

The results demonstrate that NeRP3D operates at **a computational level comparable to the voxel-based method while delivering enhanced performance**. Furthermore, we demonstrated **robust scalability, showing a clear positive correlation where detection accuracy scales with sampling density**. This behavior allows the model to utilize additional compute for maximum performance or be scaled to a balanced setting depending on the resource budget.

**3. Comparative Analysis**

- **Methodological Novelty** (Reviewer wj2C W1): We clarified that our core novelty lies not merely in the point-attention operation itself, but in the paradigm shift of establishing a **"Unified Perception Backbone" from Neural Fields** that challenges the dominant view-transformation (VT) paradigm.

- **vs. GaussianPretrain** (Reviewer 6P5j W3): We highlighted that NeRP3D achieves superior performance (+1.1 mAP) purely through volumetric learning, without the **"unfair advantage"** of explicit LiDAR initialization used by GaussianPretrain. This validates our model's capability to discover geometry from scratch.

**4. Key Design Validations & Ablation Studies**

We conducted additional ablations to verify the necessity of our architectural choices.

- **Impact of SDF Prior** (Reviewer 6P5j Q2): We verified that using SDF (NeuS) instead of density (NeRF) contributes to performance by enforcing clearer object boundaries.

- **Consistency of 3D Point Priors** (Reviewer wj2C W4): By applying NeRF pre-training to an existing point-based detector (PETRv2), we identified a "query mismatch" that leads to transfer failure. This paradoxically proves that our NeRF-resembled design, preserving the role of 3D queries, is essential for effective knowledge transfer.

- **Design Choice, Deformable Attention** (Reviewer wj2C W2a): We confirmed that deformable attention provides a necessary locality inductive bias for continuous spatial representation, outperforming standard attention.

- **Quantitative Evidence of Conflicting Prior** (Reviewer 4CMH W1): Voxel-based methods degraded significantly at high resolutions, revealing their inherent "low-pass filter" nature caused by discretization limits. In contrast, NeRP3D maintained superior fidelity with a widening performance gap at high resolutions.

We believe these additional results and analysis effectively address the reviewers' concerns and highlight the unique contributions of NeRP3D. We have incorporated these updates into the revised manuscript with appendix, highlighting all major changes in blue. We look forward to further discussion.

---

### Meta-Review · Area_Chair_zN1T · 2026-01-07

**Summary:**

Reviewers generally found the paper's systematic diagnosis of the misaligned prior between discrete view transformations and continuous radiance fields to be a valuable contribution. However, several key concerns initially hindered a consensus for acceptance:
- Reviewers `6P5j`, `wj2C`, and `4CMH` raised significant concerns regarding the lack of quantitative cost analysis (FLOPs, FPS, and memory) for the point-based architecture compared to established voxel-based baselines.
- The evaluation was initially limited to the nuScenes dataset, leading reviewers `6P5j`, `wj2C`, and `4CMH` to question the method's robustness to different sensor layouts and environmental distributions.
- Reviewer `wj2C` questioned the methodological novelty relative to existing point-querying works and noted the absence of comparisons to other point-based detectors like PETR.
- Reviewer `4CMH` requested direct quantitative evidence demonstrating that discretization actually harms performance in the context of NeRF-based pre-training.

Prior to the rebuttal, scores ranged from Marginally Below (4) to Accept (6). In the rebuttal, the authors effectively addressed these points by providing a comprehensive scalability analysis showing competitive GFLOPS and FPS, adding cross-dataset transfer results from Argoverse 2 to nuScenes, and conducting a multi-resolution analysis that proved voxel-based methods act as a low-pass filter at high resolutions. They further justified their unified architecture by showing that applying NeRF pre-training to existing point-based models like PETRv2 fails due to query mismatch.

The AC has carefully reviewed the paper, prior rebuttal reviews, rebuttals, and discussion, and agrees that the rebuttals substantially address the technical concerns, leading to a final recommendation of Accept.

**Reviewer Concerns:**

Reviewers were generally positive on the problem diagnosis and the unified continuous design (`6P5j`, `4CMH`), but raised concerns about
 (i) missing cost analysis and sampling tradeoffs (`6P5j`, `wj2C`, `4CMH`), (ii) single-dataset evaluation (`6P5j`, `wj2C`, `4CMH`), and (iii) novelty and missing baselines vs point-based detectors (`wj2C`) plus whether gains over the closest continuous baseline are meaningful (`6P5j`). The rebuttal substantively addresses cost/scale with GFLOPS/FPS/memory vs mAP comparisons, adds cross-dataset transfer (AV2→nuScenes) for detection and zero-shot rendering, provides a controlled multi-resolution reconstruction analysis supporting the “discretization as low-pass filter” narrative, and includes a diagnostic showing NeRF pre-training applied to PETRv2 fails due to query mismatch, supporting the need for a consistent point-query design. Remaining weaknesses are mostly about the breadth of external validation (no Waymo/KITTI), though the added experiments strengthen the paper’s central thesis.

**Reviewer Scores:**

Based on the rebuttal, I expect `6P5j` stays at 6 (major concerns on compute/generalization addressed), `4CMH` stays at 6 (requested evidence and cross-dataset + cost now provided), and `wj2C` likely moves from 4 to ~5–6 (key missing items of cost, ablations, cross-dataset, and the PETR comparison were addressed, though novelty/baseline breadth may still temper enthusiasm).

---

### Decision · Program_Chairs · 2026-01-26

Accept (Poster)